# Chemical genetics reveals *Leishmania* KKT2 and CRK9 kinase activity is required for cell cycle progression

Juliana B. T. Carnielli[1]*, James A. Brannigan[1,2], Priscila Z. Ramos[3], Nathaniel G. Jones[1], Rafael M. Couñago[3¤], Peter Sjö[4], Ana Paula C. A. Lima[5], Anthony J. Wilkinson[1,2], Jeremy C. Mottram[1]*

**1** York Biomedical Research Institute and Department of Biology, University of York, York, United Kingdom, **2** York Structural Biology Laboratory, York Biomedical Research Institute and Department of Chemistry, University of York, York, United Kingdom, **3** Center of Medicinal Chemistry, University of Campinas, São Paulo, Brazil, **4** Drugs for Neglected Diseases Initiative (DNDi), Geneva, Switzerland, **5** Instituto de Biofisica Carlos Chagas Filho, Universidade Federal do Rio de Janeiro, Rio de Janeiro, Brazil

¤ Current Address: Structural Genomics Consortium and Division of Chemical Biology and Medicinal Chemistry, UNC Eshelman School of Pharmacy, University of North Carolina, Chapel Hill, North Carolina, United States of America

* juliana.carnielli@york.ac.uk (JBTC); jeremy.mottram@york.ac.uk (JCM)

## Abstract

Protein kinases are key regulators of the eukaryotic cell cycle and have consequently emerged as attractive targets for drug development. Their well-defined active sites make them particularly amenable to inhibition by small molecules, underscoring their druggability. The *Leishmania* kinome, shaped by diverse evolutionary processes, harbours a unique repertoire of potential drug targets. Here, we used the cysteine-directed protein kinase probe SM1–71 to identify four essential protein kinases MPK4, MPK5, MPK7 and AEK1 as candidates for covalent kinase inhibitor development, as well as CLK1/CLK2 for which covalent inhibitors have already been identified. We leveraged the absence of natural analog-sensitive (AS) kinases in *L. mexicana* to establish an *in vivo* chemical-genetic AS kinase platform for investigating essential functions of protein kinases. Using CRISPR-Cas9-mediated precision genome editing, we endogenously engineered two kinetochore-associated protein kinases, KKT2 and KKT3, and cyclin-dependent kinase CRK9, to generate AS kinases. We show that KKT2 and CRK9 kinase activities are essential for both promastigote and intracellular amastigote survival; KKT2 kinase activity being required for progression through mitosis at a stage preceding mitotic spindle assembly, while CRK9 kinase activity is required for S phase, consistent with its role in trans-splicing. This study demonstrates the utility of AS chemical genetics in *Leishmania* and identifies KKT2 and CRK9 as having critical roles in *Leishmania* cell cycle regulation and therefore being promising drug targets.

**Data availability statement:** Mass spectrometry data sets and proteomic identifications are available to download from: MassIVE (MSV000098887 and MSV00009708), and ProteomeXchange (PXD067522 and PXD060625). All raw DNA-sequencing data generated by this study can be accessed through SRA project ID PRJNA1303394. Mapping of gatekeeper residues across the *L. mexicana* kinome, and identification of protein kinases with targetable cysteines by the multi-targeted SM1-71-biotin probe, are provided S1 and S3 Data files, respectively. Additional data generated by this study are available as supplementary information.

**Funding:** This work was supported by funding from Medical Research Council/ Global Challenges Research Fund (MRC/ GCRF - MR/P027989/1) and Wellcome Trust (223045/Z/21/Z) to JCM. The funders had no role in study design, data collection, data analysis, interpretation, decision to publish, and preparation of the manuscript.

**Competing interests:** The authors have declared that no competing interests exist.

## Author summary

In this work we aimed to better understand why some protein kinases are required for *Leishmania* parasites to grow. These parasites cause leishmaniasis, a serious disease that affects millions of people worldwide. In earlier studies, we found that some protein kinases can be targeted with drugs, but many of their specific roles remained unclear, especially in the form of the parasite that infects humans. We investigated the presence of reactive cysteine residues within the *Leishmania* kinome and identified five essential protein kinases containing cysteine residues amenable to covalent modification. We also used CRISPR-Cas9 to genetically modify *L. mexicana* protein kinases, generating analog-sensitive kinase mutants to investigate their essential functions. Our results showed that two of these kinases KKT2 and CRK9, involved in controlling the parasite's cell cycle, are indispensable for survival in the form that infects humans. These findings highlight new potential drug targets and provide a chemical genetics platform for studying how essential protein kinases function in *Leishmania*.

## Introduction

Leishmaniasis are a group of neglected tropical diseases caused by protozoan parasites of the genus *Leishmania*, transmitted to mammalian hosts by the bite of infected female sandflies. The clinical manifestations of leishmaniasis are species-specific and primarily include cutaneous leishmaniasis (CL), mucocutaneous leishmaniasis (MCL), and visceral leishmaniasis (VL). Strongly associated with poverty, leishmaniasis is endemic in 99 countries, placing over one billion people at risk and resulting in an estimated one million new cases annually worldwide [1,2].

In the absence of an effective vaccine, the limited available chemotherapies play a critical role in the control of the leishmaniasis. Currently approved treatments include pentavalent antimonials, amphotericin B, miltefosine, paromomycin, and pentamidine. However, these drugs are associated with severe shortcomings, including toxicity, the emergence of drug-resistant parasites, and restricted accessibility in low-resource settings. Moreover, the mechanisms of action of most anti-leishmanial agents, and the mechanism underlying resistance to these compounds, remain incompletely understood, impeding the rational design of combination therapies and the development of robust drug resistance surveillance strategies. To address these gaps, drug discovery efforts have increasingly leveraged phenotypic screening coupled with target deconvolution, leading to the identification of promising candidates with known mechanisms of action. Notable examples include selective inhibitors of the kinetoplastid proteasome (GNF6702 and GSK3494245) [3,4]; the pyrazolopyrimidine compound GSK3186899, which exerts its activity through inhibition of the parasite's cyclin-dependent kinase CRK12 [5]; the amidobenzimidazole AB1 from Novartis, which selectively targets the parasite kinetochore kinases CLK1/CLK2 [6,7]; the benzoxaborole derivative DNDI-6148, which acts by inhibition of *Leishmania*

cleavage and polyadenylation specificity factor (CPSF3) endonuclease [8]; and, more recently, the GSK pyrrolopyrimidine derivative, which inhibits parasite growth via targeting of mitochondrial cytochrome b [9]. Despite these advances, target deconvolution can be a major challenge, particularly when compounds act on non-protein targets or exhibit polypharmacology [10,11]. In this context, target-based drug discovery offers a complementary strategy, as it provides direct mechanistic insight and facilitates detailed structure-activity relationship studies. Nevertheless, the success of such approaches relies heavily on the availability of well-validated drug targets. For trypanosomatids, progress in this area has been limited, with only a few targets meeting established chemical and genetic criteria for target validation. Consequently, the identification of novel therapeutic targets and elucidation of their biological functions within the parasite are essential for advancing drug discovery and improving disease management.

The human kinome has been extensively explored as a therapeutic landscape, leading to the successful development of kinase inhibitors for the treatment of different types of cancers, as well as inflammatory, autoimmune, and neurodegenerative diseases [12]. This remarkable clinical impact has catalysed interest in exploring the kinome of other organisms, including protozoan parasites such as *Leishmania*, as sources of novel drug targets. The kinome of *L. mexicana*, with 193 eukaryotic protein kinases (ePKs) predicted in its genome, is roughly one-third the size of the human kinome [13,14]. Based on conserved catalytic domain sequences, ePKs are broadly classified into two major superfamilies: serine/threonine (Ser/Thr) kinases, which are ubiquitous across eukaryotes; and tyrosine (Tyr) kinases whose receptor and receptor-like members are absent in *Leishmania*. However, these organisms possess dual-specificity kinases and atypical tyrosine kinases, which likely account for protein tyrosine phosphorylation observed in their proteomes [15–18]. The *Leishmania* kinome includes homologs of the five major ePK groups found in higher eukaryotes – AGC, CAMK, CK1, CMGC, and STE – as well as members of the NEK family, which represents a distinct evolutionary branch of the human kinome. Additional kinases that do not fit into these canonical groups are categorized as "others" and include key regulators of eukaryotic cell signalling such as Aurora and Polo-like kinases, as well as the highly divergent kinetoplastid kinetochore proteins KKT2 and KKT3. Furthermore, atypical protein kinases (aPKs), which lack a conserved ePK catalytic domain but retain kinase activity, have also been identified in the *Leishmania* genome. Among these are members of the phosphatidylinositol 3′-kinase-related kinase (PIKK) group, which resemble lipid kinases [13].

Our recent functional study of the *Leishmania* kinome showed that approximately 21% of *Leishmania* protein kinases are required for promastigote survival, highlighting their potential as drug targets [14]. Notably, two preclinical candidates have provided proof-of-concept for kinase-targeted therapy in *Leishmania*: GSK3186899, a pyrazolopyrimidine that targets CRK12 [5]; and the amidobenzimidazole AB1, which selectively targets the kinetochore kinases CLK1/CLK2 [7]. These advances underscore the therapeutic promise of the *Leishmania* kinome and provide a level of validation for protein kinases as a viable class of targets for anti-leishmanial drug development.

The successful development of kinase-targeting therapeutics relies heavily on robust preclinical target validation, which involves elucidating the biological functions and signalling roles of the target protein kinase to assess its suitability as a drug target. In the absence of selective inhibitors for individual kinases, the study of kinase function has been significantly advanced by the use of the chemical-genetic analog-sensitive (AS) kinase approach. This enables precise and reversible inhibition of kinase activity. This strategy involves mutating the conserved, bulky gatekeeper residue within the ATP-binding pocket of the kinase to an amino acid with a smaller side chain, such as glycine or alanine. This substitution enlarges the ATP-binding pocket, rendering the engineered kinase uniquely sensitive to cell-permeable "bumped" kinase inhibitors (BKIs), structurally modified ATP analogs that do not bind to the wild-type kinase ATP-binding pocket, but can selectively inhibit the analog-sensitive variant [19–21].

AS kinase technology has been successfully applied to investigate protein kinase signalling in diverse cellular processes such as cell cycle [22,23], immunological T and B cell activation [24,25], and apicomplexan parasite egress [26]. This approach was envisioned for kinetoplastid parasites two decades ago [27] and subsequently brought to light the biological role of the cyclin-dependent kinases CRK1 and CRK9 in *Trypanosoma brucei*. CRK1 has been shown to be

involved in anterograde protein trafficking [28], and to promote the G1/S phase transition of the cell cycle, in coordination with global protein translation [29], while CRK9 is essential for pre-mRNA processing [30]. Additionally, AS-mediated inhibition of polo-like kinase has revealed its critical role in flagellar attachment and cytokinesis in *T. brucei* [31]. Importantly, this chemical-genetic strategy has also contributed to drug discovery by validating AGC essential kinase 1 (AEK1) as a potential drug target in *T. brucei* [32] and *T. cruzi* [33].

Equally important to the identification of potential drug targets is the ability to rationally design selective inhibitors against them. In this context, substantial progress has been achieved through the development of targeted covalent kinase inhibitors. These inhibitors are typically engineered by combining a kinase-binding scaffold with an electrophilic warhead capable of forming an irreversible covalent bond with nucleophilic amino acid residues [34]. The most commonly targeted residue is cysteine, due to its high intrinsic nucleophilicity [35]. Notably, the eleven FDA-approved covalent kinase inhibitors operate through a shared mechanism involving the Michael addition of a protein cysteine thiolate to an acrylamide or acrylamide-like electrophile [36]. This strategy has also shown promise against trypanosomatid parasites, where CLK1/CLK2 inhibition by AB1 illustrates the potential of covalent kinase inhibitors in the context of trypanosomatid drug discovery [6]. Similarly, in *Plasmodium falciparum*, Cys368 of the essential cyclin-dependent-like protein kinase 3 (*Pf*CLK3) was successfully targeted by a covalent chloroacetamide inhibitor [37]. Therefore, the identification of reactive cysteine residues within protein kinases has emerged as a valuable approach to guide rational covalent inhibitor design. One powerful tool in this effort has been the promiscuous covalent kinase inhibitor SM1–71, originally developed to target cysteines located immediately N-terminal to the conserved DFG motif (DFG-1) in human protein kinases. The use of SM1–71 and its biotinylated analog has enabled systematic mapping of reactive cysteine residues across the human kinome, providing key insights into targetable nucleophilic sites for covalent ligand development [38,39].

In this study, we explored the active sites of the *Leishmania* kinome using the SM1–71 probe. We identified 10 kinases with targetable cysteine residues, five of which are known to be essential for parasite survival. Our bioinformatic and structural analysis indicate an absence of naturally occurring AS kinases that have glycine or alanine gatekeeper residues, making this parasite a suitable model for the AS chemical genetics strategy in the promastigote and, importantly, the intracellular amastigote stages. We showed that BKI-mediated inhibition of KKT2 AS resulted in mitotic arrest at a stage prior to spindle assembly, whilst inhibition of CRK9 AS led to impaired DNA synthesis, resulting in cell cycle arrest.

## Results

### *Leishmania* kinome, a source of attractive drug targets

From our previous systematic analysis of the *Leishmania* kinome, 43 protein kinases were highlighted as potentially being essential in the promastigote stage, since a null mutant could not be recovered [14]. In addition, we have shown that the kinetochore protein kinases CLK1 and CLK2, whose kinase domains share 100% sequence identity, exhibit biological redundancy, and that simultaneous inhibition of both enzymes leads to loss of promastigote viability [7,40]. Consequently, CLK1 and CLK2 were also classified as required, bringing the total number of protein kinases required for promastigote survival to 45 (S1 Table). Here, we reanalysed the Bar-Seq dataset generated from the pooled barcoded library of kinase deletion mutants, to identify protein kinases required for amastigote-stage survival. The proportion of barcodes in the metacyclic promastigote stage, as well as at two post-infection time points – in *in vitro* macrophage (12 and 72 hours) and in *in vivo* mouse footpad (3 and 6 weeks) experiments – was used to calculate fold changes in barcode representation relative to the preceding time point. A protein kinase was classified as required for amastigote survival if the corresponding mutant cell line exhibited the following trajectory in barcode representation: (i) a significant ≥50% reduction in barcode abundance at the first post-infection time point, followed by an additional decrease of ≥30% at the second time point; or (ii) no significant change in the first time point, but a significant ≥50% reduction at the final time point (S1 Fig). This scoring approach was expected to exclude kinases primarily involved in parasite infection or differentiation. This analysis

identified 18 protein kinases required for amastigote survival. These proteins were distributed across kinase families as follows: AGC (1), CAMK (2), CK1 (2), CMGC (7), NEK (2), STE (1), and Other (3) (Table 1).

From our comprehensive analysis of the kinase requirements in *Leishmania*, 18 protein kinases were identified as essential for the survival of the amastigote stage, in addition to the 45 previously shown to be required in promastigotes [14] (S1 Table). This brings the total to 63 protein kinases that represent potential drug targets. Among these, particular attention was drawn to the four Kinetoplastid Kinetochore Proteins (KKTs) – CLK1 (LmxM.09.0400, also annotated as KKT10), CLK2 (LmxM.09.0410, also annotated as KKT19), KKT2 (LmxM.36.5350), and KKT3 (LmxM.34.4050) – due to their divergent evolutionary roots in the conserved eukaryotic chromosome segregation process, resulting in a unique and druggable repertoire specific to trypanosomatids. We have previously validated CLK1 and/or CLK2 as druggable targets in trypanosomatids through the development of AB1 [6,7,41] and the third-generation EGFR inhibitor WZ8040, using a bioluminescence resonance energy transfer (BRET)-based target engagement assay in live cells [40].

### *L. mexicana* protein kinase function investigated using chemical-genetics

Given that the most robust target validation strategies enable assessment of drug effects *in vivo*, we sought to engineer analog-sensitive kinases directly in *Leishmania*. Genetic modification of the gatekeeper residue was introduced in the amenable promastigote stage, followed by differentiation into intracellular amastigotes – the clinically relevant, drug-targetable form in the mammalian host. This strategy allows evaluation of the functional requirement of the engineered kinases across both life stages using susceptibility assays. If the target kinase activity is essential, AS kinase-expressing parasites would be expected to exhibit increased susceptibility to BKIs relative to the wild-type T7/Cas9 line.

To assess the feasibility of using AS kinase technology to investigate the requirement of protein kinases for *Leishmania* survival, we first screened the *L. mexicana* kinome for the presence of naturally occurring AS kinases that might interfere

**Table 1. *L. mexicana* protein kinases required for amastigote survival.**

| Gene ID | Name | Group/Family | Amastigote survival requirement | |
|---|---|---|---|---|
| | | | inMAC [a] | FP [b] |
| LmxM.28.1670 | ZFK | AGC | **Required** | Dispensable |
| LmxM.19.1470 | | CAMK/CAMKL | Dispensable | **Required** |
| LmxM.22.0810 | SOS2 | CAMK | Dispensable | **Required** |
| LmxM.04.1210 | CK1.3 | CK1/CK1 | Dispensable | **Required** |
| LmxM.33.3020 | | CK1/TTBK | **Required** | Dispensable |
| LmxM.13.1640 | MPK7 | CMGC/MAPK | Dispensable | **Required** |
| LmxM.20_36.6470 | MPK1 | CMGC/CDKL | **Required** | Dispensable |
| LmxM.21.1650 | PK4 | CMGC | Dispensable | **Required** |
| LmxM.25.1560 | | CMGC/CLK | Dispensable | **Required** |
| LmxM.29.0370 | MPK12 | CMGC/MAPK | Dispensable | **Required** |
| LmxM.29.2910 | MPK5 | CMGC/MAPK | Dispensable | **Required** |
| LmxM.36.4250 | | CMGC/GSK | Dispensable | **Required** |
| LmxM.10.0830 | | Other/Orphan | Dispensable | **Required** |
| LmxM.33.0030 | | Other/NAK | Dispensable | **Required** |
| LmxM.33.0940 | WEE1 | Other/WEE | Dispensable | **Required** |
| LmxM.08_29.2570 | | NEK | Dispensable | **Required** |
| LmxM.30.3160 | | NEK | Dispensable | **Required** |
| LmxM.07.0250 | | STE | Dispensable | **Required** |

[a] inMAC, *in vitro* infected macrophage. [b] FP, *in vivo* mouse footpads.

with our approach. A natural AS kinase is defined as a kinase that has a glycine or alanine residue in the gatekeeper position. To address this, we performed a comprehensive protein sequence and structural analysis of all 183 *bona fide* ePKs described in the *L. mexicana* kinome to determine the identity of their gatekeeper residues. Of the 193 predicted ePKs previously annotated in the *L. mexicana* kinome [14], 10 are classified as pseudokinases as they lack all three of the conserved catalytic residues (K D D) required for kinase activity (Fig 1a). Given their lack of enzymatic function and demonstrated non-essentiality, pseudokinases were not analysed for gatekeeper residue identity.

Our analysis revealed that methionine is the predominant gatekeeper residue, present in 105 of the 183 genuine ePKs (57.4%). Leucine (19.7%), phenylalanine (10.4%), and threonine (8.2%) were the next most abundant gatekeeper residues. Notably, the AGC kinase subgroup was the only family in which methionine was not the most frequent gatekeeper; instead, leucine was most prevalent (63.6%). The two smallest gatekeeper residues identified were: a cysteine in the member of the family STE LmxM.36.0910; and a serine in LmxM.36.4250, a member of the CMGC family (Figs 1b and S2 and S1 Data).

We also analysed the gatekeeper residues of 23 atypical protein kinases. Interestingly, valine was the most frequent gatekeeper residue among aPKs, a residue not observed at this position in any ePK (Figs 1c and S2 and S1 Data). However, due to their distinct structural features, aPKs are inhibited by a different set of BKIs than those used for analog-sensitive ePKs (e.g., LY294002 analogues targeting PI3K-like kinases) [42].

Together, these findings demonstrate that *L. mexicana* lacks natural analog-sensitive kinases, supporting the suitability of this parasite as a model for functional dissection of protein kinase activity using AS technology.

We selected the kinetochore kinases CLK1/CLK2, KKT2 and KKT3 for functional validation using the chemical-genetic AS kinase strategy. Additionally, we have also selected the cyclin-dependent kinase CRK9 (LmxM.27.1940) to target, since this essential protein kinase was shown to be amenable to the AS kinase approach in the closely related trypanosomatid *T. brucei* [30]. Orthologs of *L. mexicana* CLK1/CLK2, KKT2, KKT3 and CRK9 are present and highly conserved in all high-quality sequenced genomes of the order *Trypanosomatida* that are known human pathogens (S3 Fig), making studies on *Leishmania* relevant to multiple clinically relevant trypanosomatid parasites. Our gatekeeper residue analysis revealed that a bulky methionine occupies the gatekeeper position of the ATP-binding site of these five selected kinase targets (Fig 2a).

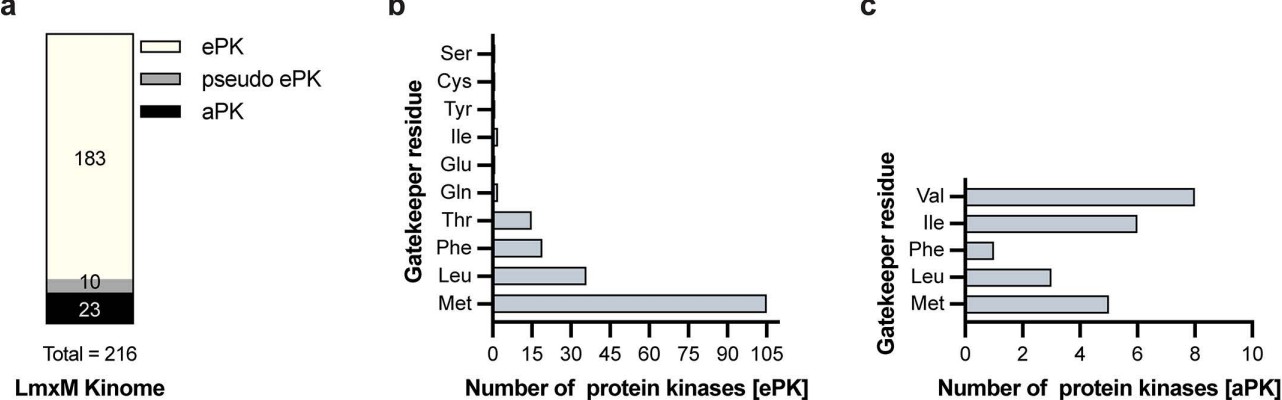

**Fig 1. The *L. mexicana* kinome lacks natural analog-sensitive kinases. (a)** Schematic overview of the *L. mexicana* kinome, comprising eukaryotic protein kinases (ePKs) and atypical protein kinases (aPKs). ePKs lacking all three key residues (K D D) required for catalytic activity were classified as pseudo ePKs. **(b)** Distribution of gatekeeper residues among the 183 *bona fide L. mexicana* ePKs. **(c)** Gatekeeper residue distribution among the 23 *L. mexicana* aPKs.

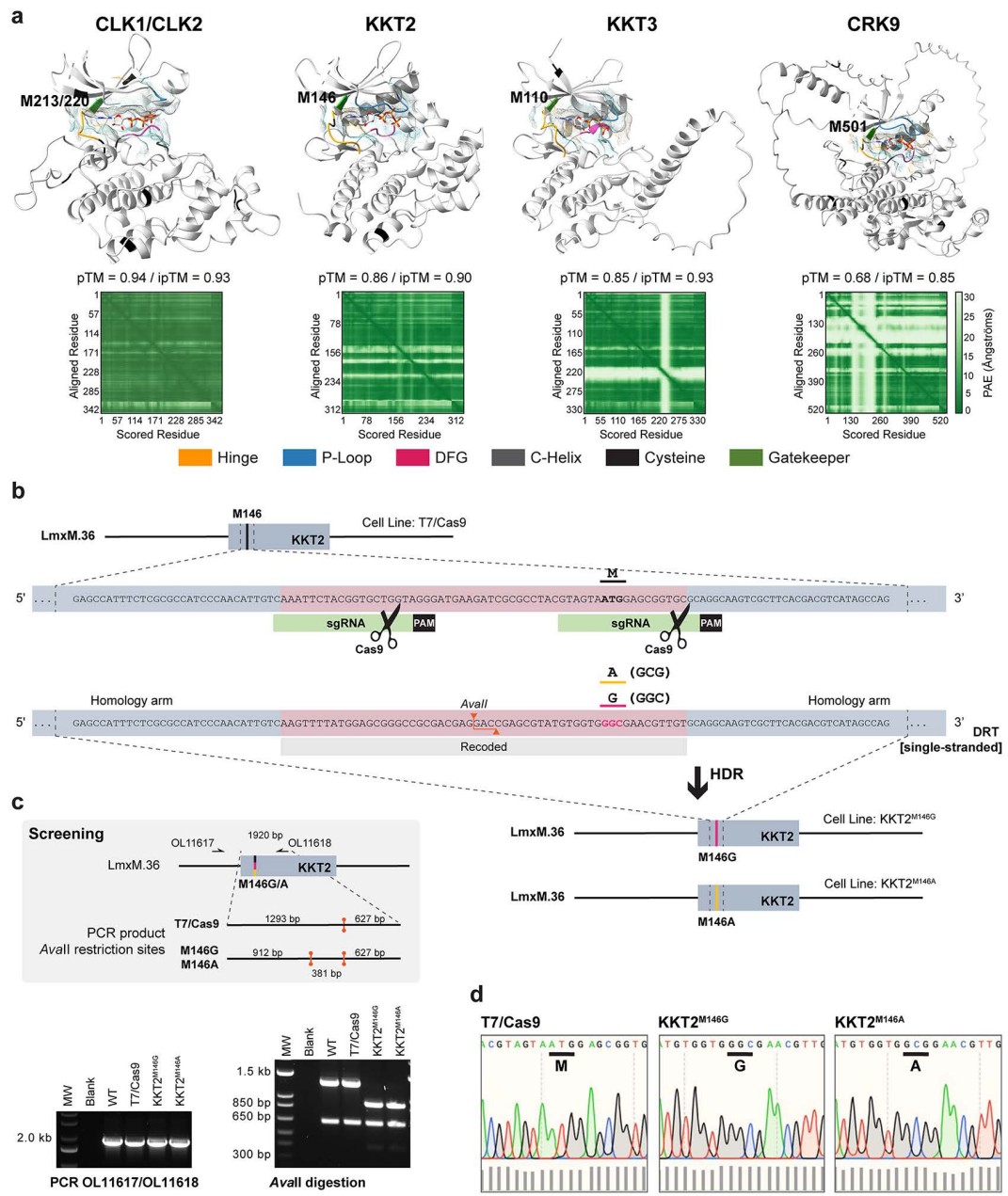

**Fig 2. Marker-free CRISPR-Cas9-mediated precision genome editing to engineer analog-sensitive kinases in *Leishmania*. (a)** Structural model of the kinase domains of CLK1/CLK2, KKT2, KKT3, and CRK9 generated using AlphaFold 3 [43] and visualized with ChimeraX v1.9. The ATP ligand is displayed as a stick model, with heteroatom-based colouring. A semi-transparent molecular lipophilicity potential (MLP) surface is overlaid, with colour ranging from dark cyan (most hydrophilic) to white to dark goldenrod (most lipophilic). AlphaFold confidence metrics are displayed below each structure, including the predicted template modelling score (pTM), the interface predicted template modelling score (ipTM) between the kinase and ATP, and the predicted aligned error (PAE) plot. **(b)** Schematic representation of the CRISPR-Cas9 strategy used to engineer analog-sensitive kinases by substituting the gatekeeper methionine (M) with glycine (G) or alanine **(A)**. The example illustrates the replacement of the gatekeeper residue in KKT2. Linear DNA fragments for *in vivo* transcription of two single guide RNAs (sgRNAs) and a 120 bp DNA repair template containing silent recoding mutations and the gatekeeper substitution were used. The mutations introduced an *Ava*II restriction site, enabling genotypic screening of edited clones. PAM, protospacer adjacent motif; HDR, homology-directed repair; LmxM.36, *L. mexicana* chromosome 36. **(c)** Genotyping workflow (top grey box) and PCR-restriction digest results (bottom) for selected analog-sensitive clones. **(d)** Sanger DNA sequencing of the engineered KKT2 locus confirms the substitution of the gatekeeper methionine with glycine or alanine in the KKT2^M146G and KKT2^M146A lines, respectively. Sequencing chromatograms were visualized in Snap-Gene v7.2; bar graphs below indicate per-base quality scores.

## CRISPR-Cas9-mediated precision genome editing to engineer *Leishmania* protein kinases

The *L. mexicana* cell line stably expressing T7 RNA polymerase and Cas9 endonuclease [44] was used as the parental strain for engineering AS protein kinases via CRISPR-Cas9-mediated precision genome editing. Analog-sensitive kinase mutants were generated by introducing specific point mutations at the endogenous kinase *loci*, replacing the native bulky gatekeeper residue with either glycine or alanine. This was achieved using a 120 bp single-stranded DNA repair template lacking selectable drug resistance markers. To direct the desired edits, two single guide RNAs (sgRNAs) were designed to flank the gatekeeper codon. In addition to the codon substitution, the DNA repair template was engineered to contain silent mutations within the protospacer adjacent motif (PAM) and sgRNA target sites to prevent its cleavage by Cas9 (Fig 2b). These silent mutations also introduced/removed restriction endonuclease cleavage sites, enabling discrimination between wild-type and mutant alleles for downstream screening. Ten clones per transfection were screened by PCR followed by restriction enzyme digestion and then confirming by Sanger DNA sequencing (Fig 2c–2d). This strategy was successful in the generation of AS kinase mutants for KKT2, KKT3 and CRK9 with 3/10 success rate for the KKT2$^{M146G}$ substitution, 1/10 for KKT2$^{M146A}$, 1/10 for KKT3$^{M110G}$, and 1/10 for both CRK9$^{M501G}$ and CRK9$^{M501A}$ variants (Figs 2b–2d and S4–S6). These findings demonstrate that *L. mexicana* is amenable to marker-free precision genome editing using the CRISPR-Cas9 system, with high editing efficiency achieved across multiple protein kinase targets.

CLK1 and CLK2 share 86% sequence identity, and their kinase domains, which comprise the gatekeeper residue (M213 in CLK1 and M220 in CLK2), are identical (S7a Fig). As a result, it is not feasible to selectively target CLK1 without also affecting CLK2, and *vice versa*. Therefore, both kinases were simultaneously targeted in this study. Nevertheless, precision editing of CLK1 or CLK2 was not achieved (S7b–S7f Fig).

To investigate whether the gatekeeper mutations affect CLK1 catalytic activity, recombinant wild-type CLK1 (rCLK1) and its analog-sensitive variants (rCLK1$^{M213A}$ and rCLK1$^{M213G}$) were expressed and purified (Figs 3a and S8a–S8b). Kinase activity was assessed using a homogenous time-resolved fluorescence (HTRF)-based assay, previously validated for *L. mexicana* rCLK1 [40]. Phosphorylation of the substrate was detected for rCLK1, whereas no detectable activity was observed for either rCLK1$^{M213A}$ or rCLK1$^{M213G}$ (Fig 3b). To further examine substrate recognition, four additional peptide substrates were tested using a time-resolved fluorescence resonance energy transfer (TR-FRET)-based assay: CREBtide (derived from human cAMP Response Element Binding (CREB) protein), Histone H3 (Thr3) (derived from human RAC-alpha serine/threonine-protein kinase), 4E-BP1 (derived from human eukaryotic translation initiation factor 4E-binding protein 1), and GS (derived from human Muscle Glycogen Synthase). rCLK1 efficiently phosphorylated CREBtide and GS, whereas neither analog-sensitive variant displayed detectable activity toward any of the peptides tested (S8c Fig). Auto-phosphorylation was next evaluated by intact mass spectrometry. Following incubation with ATP, rCLK1 exhibited three distinct phosphorylation states, consistent with robust autophosphorylation. In contrast, no phosphorylation was detected for the rCLK1$^{M213G}$ variant, and the rCLK1$^{M213A}$ variant displayed a single phosphorylation state at very low abundance (Fig 3c), indicating markedly impaired catalytic activity. In addition, differential scanning fluorimetry revealed distinct thermal unfolding profiles for rCLK1 compared with the analog-sensitive variants, consistent with altered protein stability upon gatekeeper substitution (Fig 3d). Taken together, these findings suggest that the inability to generate CLK1/CLK2 analog-sensitive mutants in *Leishmania* is due to a substantial impairment of essential kinase activity caused by the gate-keeper mutation.

## KKT2 and CRK9 protein kinase activity is essential for *Leishmania*

The requirement for kinase activity was initially evaluated in the promastigote stage of the *Leishmania* parasite by comparing the sensitivity of parasites expressing either wild-type or analog-sensitive kinases to the bumped kinase inhibitors 1NM-PP1 and 1NA-PP1 (Figs 4a and S9a). Using a resazurin-based assay, the effects of the tested BKIs on *Leishmania* promastigote metabolic state revealed that 1NM-PP1 exhibited stronger inhibitory activity, partic-ularly against AS kinases with a glycine residue at the gatekeeper position, suggesting more efficient binding in

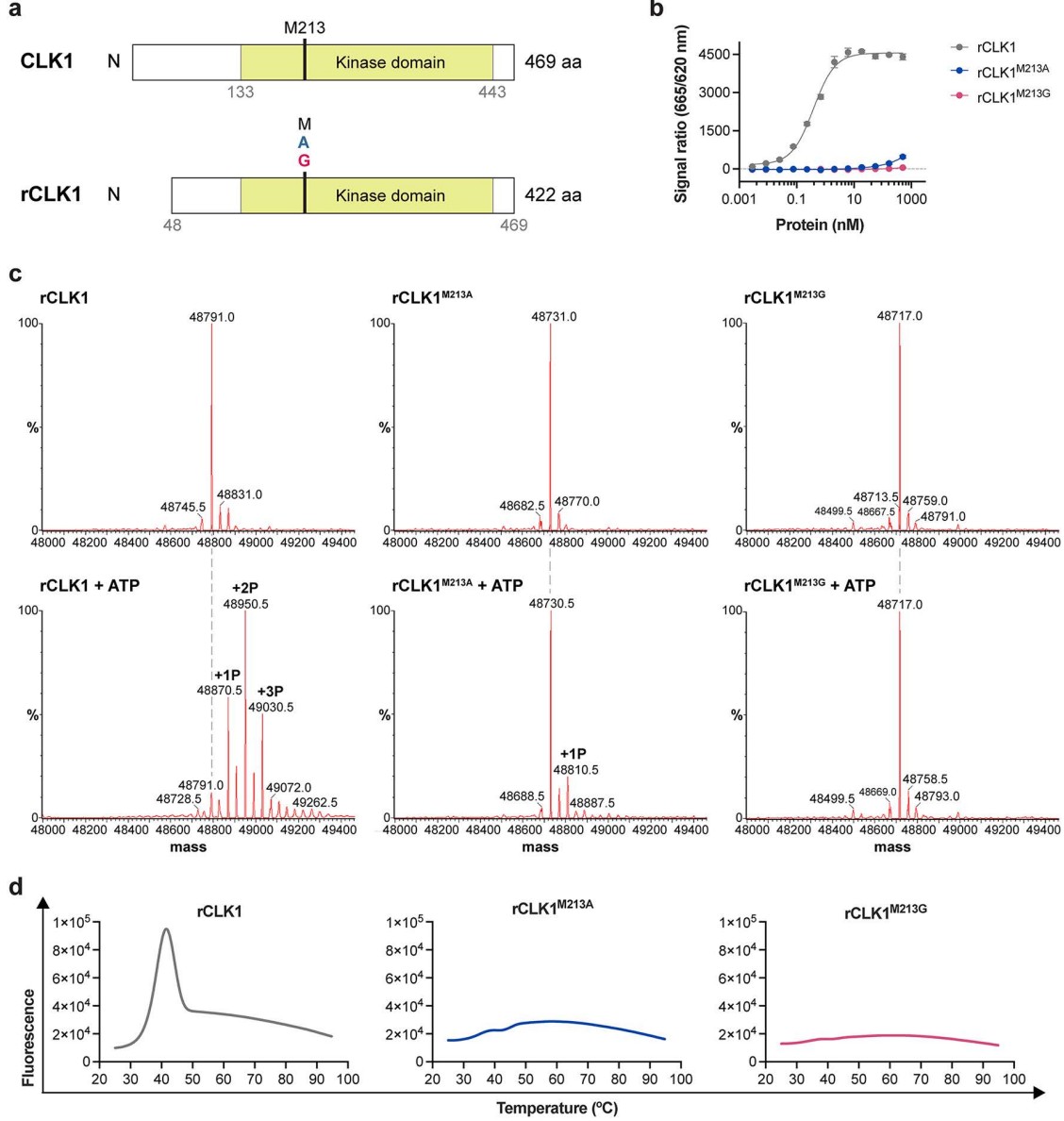

**Fig 3. Gatekeeper substitution impairs CLK1 kinase activity. (a)** Schematic representation of full-length CLK1, including the previously predicted N-terminally extended coding sequence [45], and the recombinant protein. **(b)** Kinase activity of recombinant CLK1 (rCLK1) and its analog-sensitive variants (rCLK1$^{M213A}$ and rCLK1$^{M213G}$) measured using the serine/threonine kinase STK-S3 assay. Concentration-dependent phosphorylation of the peptide substrate is shown. **(c)** Autophosphorylation activity assessed by liquid chromatography-mass spectrometry (LC-MS). Intact mass spectra of purified proteins before (top panels) and after incubation with 1 mM ATP for 2 h at 37 °C (bottom panels) are shown. Deconvoluted spectra are presented as molecular mass (Da). The expected intact masses of rCLK1, rCLK1$^{M213A}$ and rCLK1$^{M213G}$ are 48,791.1, 48,731.0 and 48,717.0 Da, respectively. Phosphorylation was inferred from mass shifts of +80.0 Da per phosphate group. **(d)** Differential scanning fluorimetry thermal melting curves of rCLK1 and its analog-sensitive variants.

these mutants. On the other hand, 1NA-PP1 displayed reduced specificity, showing comparable inhibition of AS kinases with either glycine or alanine at the gatekeeper site. Parasites expressing AS variants of KKT2 and CRK9 exhibited marked growth inhibition at BKI concentrations that had minimal effect on the parental line (T7/Cas9 IC$_{10}$).

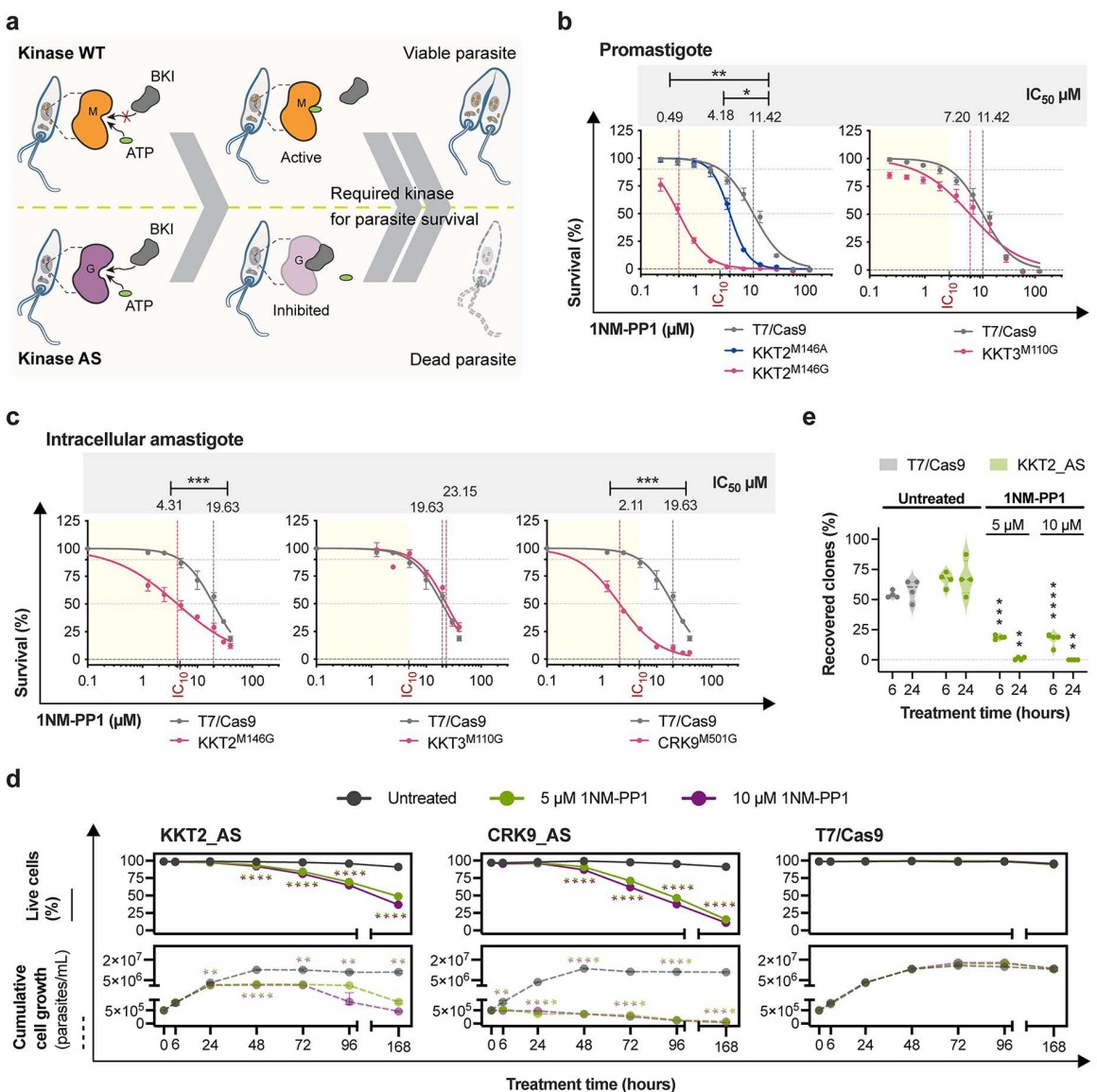

**Fig 4. The kinase activities of KKT2 and CRK9 are essential for *Leishmania* survival. (a)** Schematic representation of the viability assay performed on parasites expressing either the wild-type or AS variant, illustrating the expected outcome if the targeted protein kinase is essential for parasite survival following treatment with a bumped kinase inhibitor. The same experimental principle was applied to the amastigote susceptibility assay. WT, wild type; AS, analog-sensitive; M, methionine; G, glycine; BKI, bumped kinase inhibitor. Parasite susceptibility to 1NM-PP1 treatment was assessed in *L. mexicana* lines expressing AS variants of KKT2, KKT3, and CRK9, as well as the parental T7/Cas9 line. Dose-response curves were fitted using GraphPad Prism v10.4.1, with survival normalized to untreated controls (set at 100% for each cell line). Statistical significance was evaluated using unpaired two-tailed Student's t-tests (*p-value <0.05; **p-value <0.01; ***p-value <0.001). The $IC_{10}$ value denotes the concentration of 1NM-PP1 that reduces the viability of the parental T7/Cas9 line by 10%. **(b)** Susceptibility of promastigotes to 1NM-PP1 was measured using a resazurin-based assay. Data represent mean±SEM from three biological replicates. **(c)** Susceptibility of intracellular amastigotes to 1NM-PP1 was assessed by quantifying the percentage of infected macrophages. Data represent mean±SEM from four biological replicates. **(d)** Effects of selective kinase inhibition on cell viability and proliferation in AS variants of KKT2 and CRK9. The parental T7/Cas9 line was used as a control. Cell viability was assessed by propidium iodide staining and quantified by flow cytometry (top panel). Cumulative cell densities were determined by manual counting using a Neubauer chamber (bottom panel). Data represent mean±SEM from four biological replicates. Statistical comparisons between treated and the untreated samples were performed using unpaired two-tailed Student's t-test (* p-value <0.05; ** p-value <0.01; *** p-value <0.001; **** p-value <0.0001). Proliferation was compared on the basis of calculated growth rates. **(e)** Washout clone recovery following KKT2 AS inhibition. The percentage of recovered clones was quantified after 6 or 24 hours of KKT2 AS inhibition with 1NM-PP1, followed by compound washout. Untreated KKT2 AS and T7/Cas9 lines were used as controls. Data represent mean±SEM from four biological replicates. Statistical comparisons were performed relative to untreated KKT2 AS samples using unpaired two-tailed Student's t-tests (** p-value <0.01; *** p-value <0.001; **** p-value <0.0001).

Furthermore, the AS mutants exhibited significantly increased sensitivity to BKIs, as indicated by lower half-maximal inhibitory concentration ($IC_{50}$) values compared to the parental line. Our result on the sensitivity of CRK9 AS to BKIs in promastigote stage was previously published by Jones et al., 2023 [46]. Using the dataset generated in our previous publication, we calculated $IC_{50}$ values and compared them to those of the parental line T7/Cas9, which displayed $IC_{50}$ values of 11.26 μM for 1NM-PP1 and 12.80 μM for 1NA-PP1. The CRK9$^{M501A}$ showed reduced $IC_{50}$ values of 6.22 μM for 1NM-PP1 (p-value = 0.0469) and 3.36 μM for 1NA-PP1 (p-value = 0.0469), while the CRK9$^{M501G}$ exhibited even greater sensitivity, with $IC_{50}$s of 0.63 μM for 1NM-PP1 (p-value = 0.0071) and 2.32 μM for 1NA-PP1 (p-value = 0.0231). These findings indicate that the kinase activity of KKT2 and CRK9 is essential for promastigote survival (Figs 4b and S9b, and Jones et al., 2023 [46]).

Although the AS kinase mutants retained sufficient activity to support promastigote growth in culture, we assessed whether the introduced mutations affected parasite fitness. Growth rate comparisons between AS lines carrying glycine gatekeeper mutations and the parental line revealed no significant differences (S10 Fig), confirming that the increased drug sensitivity observed in viability assays was due to specific inhibition of kinase activity, rather than a fitness defect.

Given the increased susceptibility of parasites expressing analog-sensitive kinases with a glycine gatekeeper to BKIs, these lines were selected for intracellular amastigote susceptibility assays. Consistent with observations in the promastigote stage, parasites expressing AS variants of KKT2 and CRK9 showed significantly increased susceptibility to 1NM-PP1, indicating that the kinase activity of these proteins is also critical for survival of the intracellular amastigote stage (Fig 4c). Notably, 1NA-PP1 exhibited substantially lower activity against the intracellular amastigote stage compared to 1NM-PP1 (Figs 4c and S9c). Importantly, at the end of the intracellular amastigote susceptibility assay under untreated conditions, the analog-sensitive kinase lines displayed percentages of infected macrophages comparable to those of the parental T7/Cas9 control, indicating that the gatekeeper mutation did not impair parasite infectivity and/or proliferative fitness (S9d Fig). To evaluate potential host-cell toxicity, we assessed the effect of 1NM-PP1 on bone marrow-derived macrophages (BMDM), the host cell used for intracellular infection assays. 1NM-PP1 exhibited significant cytotoxicity only at concentrations ≥40 μM (48.6% reduction in cell metabolic activity), which corresponded to the highest concentration tested (S9e Fig).

To further evaluate KKT2 and CRK9 as potential drug targets, we took advantage of selective chemical inhibition to determine whether loss of kinase activity affects cell viability or proliferation. As both kinases are predicted to participate in cell-cycle regulation [46,47], we first examined the early effects of kinase inhibition. To this end, cells were treated for 6 h, as the duration of a complete cell cycle in *L. mexicana* promastigotes has been estimated at approximately 7.1 h [48]. Under these conditions, most asynchronously growing cells are expected to progress through a substantial fraction of the cell cycle, enabling detection of cell-cycle arrest or delay upon kinase inhibition. Samples were subsequently analysed at 24 h intervals to assess the consequences of prolonged inhibition. Inhibitor concentrations were selected based on the 96 h susceptibility assays. Concentrations of 5 μM and 10 μM 1NM-PP1 were chosen and inhibition of CRK9 and KKT2 initially resulted in a significant reduction in cell proliferation after 6 and 24 h, respectively. Loss of viability was observed after 48 h of inhibition, indicating a primarily cytostatic effect and that sustained inhibition ultimately leads to parasite death. Consistent with a time-dependent effect, most cells lost viability after one week of treatment. In the KKT2 AS line, 51.3% and 63.2% of cells were non-viable following treatment with 5 μM and 10 μM 1NM-PP1, respectively, whereas in the CRK9 AS line the corresponding proportions of non-viable cells were 84.3% and 89%. Importantly, these conditions did not impair growth or viability of the parental T7/Cas9 control line (Fig 4d).

To determine whether cells arrested upon KKT2 AS inhibition were capable of recovery, we performed washout experiments in which the inhibitor was removed after 6 or 24 h of treatment, and clone recovery was quantified relative to untreated controls. Even after 6 h of inhibition, the proportion of recovered clones was significantly reduced. Following

24 h of treatment, only 1.05% of clones recovered after exposure to 5 µM 1NM-PP1, and no recovery was observed at 10 µM (Fig 4e). These findings demonstrate that KKT2 inhibition rapidly compromises parasite proliferative capacity and that even short-term inhibition markedly reduces the ability of cells to recover, with prolonged inhibition resulting in an essentially irreversible loss of proliferative potential.

### Failure of the analogue-sensitive KKT3 approach, although KKT3 kinase activity appears essential for *Leishmania* viability

The susceptibility of parasites expressing AS variant of KKT3 to the BKIs did not differ from that of the parental line in either the promastigote or intracellular amastigote stages (Figs 4b–4c and S9b–S9c), initially suggesting that KKT3 kinase activity could not be essential for parasite survival. To reassess the essentiality of *KKT3*, we attempted CRISPR-Cas9-mediated gene deletion using single guide RNAs (sgRNAs) designed with the updated LeishGEdit algorithm for Bar-Seq applications [49]. Puromycin and blasticidin double-resistant parasites were recovered in two of three independent transfections. However, PCR analysis revealed that the KKT3 coding sequence was still present in recovered clones, suggesting that the gene or chromosome may have been duplicated to retain an essential function. Whole-genome sequencing of three recovered clones confirmed gene duplication of the *KKT3* locus, supporting the hypothesis that KKT3 is essential for promastigote viability (S11 Fig).

To further investigate whether KKT3 kinase activity is required for parasite survival, we attempted to generate catalytically inactive (kinase-dead) KKT3 mutants by substituting the three conserved catalytic residues (K64, D156, and D174) with alanine using precision editing. This approach was planned to first introduce simultaneous mutations at D156 and D174, followed by targeting K64 (Fig 5a). To achieve this, we employed our drug-free CRISPR-Cas9 system with a double-stranded DNA repair template, which we previously demonstrated to be more efficient than single-stranded templates [50,51]. Repair templates carrying identical recoded codons but silent substitutions at the target residues were used as controls. Initial screening yielded clones consistent with correct integration of the repair templates at both KKT3 alleles.

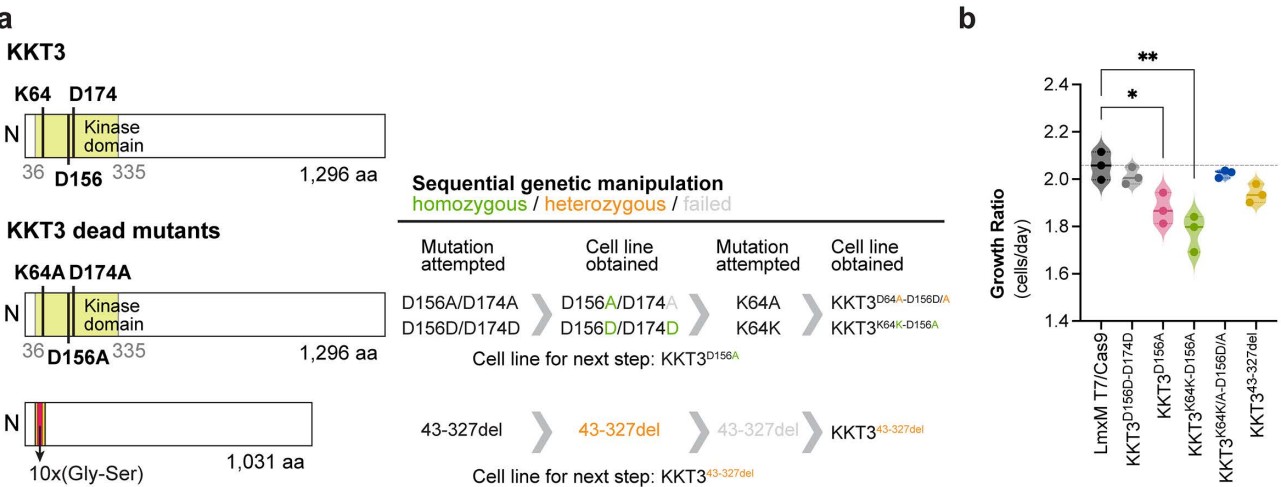

**Fig 5. Evidence that KKT3 kinase activity is required for *Leishmania* survival. (a)** Schematic overview of the strategy employed to generate kinase-dead mutants of KKT3. The left panel illustrates the wild-type KKT3 protein and the corresponding kinase-inactive variants. The right panel depicts the sequential genetic manipulations performed and the resulting cell lines obtained at each step. **(b)** Proliferation analysis of kinase-dead mutants. Growth rates were calculated from the logarithmic phase of the growth curve (0 – 72 h) and are reported as mean ± SEM of three biological replicates. Statistical comparisons between mutants and the parental T7/Cas9 line were performed using unpaired two-tailed Student's t-test (* $p < 0.05$; ** $p < 0.01$).

However, sequencing analysis revealed that, whereas the control line successfully incorporated the complete repair template, substitution of the DFG aspartate (D174A) could not be recovered, indicating that this mutation is not tolerated. We then proceeded with the mutant line in which the catalytic aspartate (D156) was replaced by alanine (KKT3$^{D156A}$) and subsequently attempted to mutate the conserved lysine (K64) within the N-lobe β-strand. Although control edits confirmed that this genomic region is amenable to manipulation, the only clone recovered carrying the K64A substitution was heterozygous. Interestingly, sequencing of this clone revealed reversion of the previously mutated region to the wild-type sequence in one allele, including restoration of the catalytic aspartate D156 (S12 Fig).

In parallel, using the same CRISPR-Cas9 strategy, we aimed to delete the KKT3 kinase domain by replacing residues 43 – 327 with a flexible 10x glycine-serine linker (Fig 5a). This approach yielded a single heterozygous clone, which was subsequently subjected to a second round of transfection with the same repair template, however, no homozygous mutants were recovered (S13 Fig).

We next assessed the fitness of the recovered mutant lines in culture relative to the parental T7/Cas9 line. Introduction of silent mutations at KKT3 D156 and D174 had no detectable effect on parasite growth. In contrast, the KKT3$^{D156A}$ mutant displayed a significantly reduced growth rate during the logarithmic phase. This growth defect persisted following introduction of the silent K64 mutation but was no longer observed in the KKT3$^{K64K/A-D156D/A}$ heterozygous line. Parasites carrying a heterozygous deletion of the KKT3 kinase domain exhibited a non-significant trend towards reduced growth (Figs 5b and S14). Altogether, these findings suggest that KKT3 kinase activity is essential for parasite survival and that gatekeeper substitution alone does not sensitize KKT3 to inhibition by 1NM-PP1 or 1NA-PP1.

In *T. brucei* KKT3, in addition to the kinase domain, a centromere localization domain and a divergent polo box domain have been identified, with the polo box shown to be essential for parasite survival [52]. Protein sequence and structural analyses revealed that *L. mexicana* KKT3 similarly harbours a centromere localization domain and a divergent polo box domain (S15 Fig). However, further investigation is required to assess the contribution of these additional KKT3 domains to *Leishmania* parasite survival.

## KKT2 kinase activity is required for coordinated progression through S and M phases in *Leishmania*

The effect of KKT2 inhibition on cell cycle progression in *Leishmania* promastigotes was assessed by measuring the DNA content via flow cytometry. Cells were stained with propidium iodide (PI), and its distribution across cell cycle phases was determined using the Watson model algorithm in FlowJo v10.10.0. Following treatment with 5 μM 1NM-PP1 for 6 hours, parasites expressing the KKT2$^{M146G}$ analog-sensitive variant exhibited a significant accumulation of cells in the G2/Mitosis phase (30.6%), compared to the untreated KKT2 AS control (21.5%) and wild-type KKT2 parasites subjected to the same treatment (21.3%), indicating cell cycle arrest at this stage (Figs 6a and S16a). Concomitantly, treated KKT2 AS parasites displayed a reduction in the proportion of cells in S phase and in multinucleated cells (MNC) relative to treated wild-type parasites, as well as a decrease in the G1 population when compared with untreated KKT2 AS controls. After 24 hours of treatment, a further reduction in the proportion of cells in the G1 phase was detected in the treated parasites expressing the KKT2 AS variant. Moreover, comparison of treated *versus* untreated AS parasites revealed an accumulation of cells in S phase upon KKT2 inhibition. While the Watson model predominantly classified these cells as being in S phase, visual inspection of the histograms indicated that the G2/Mitosis population was also enlarged. Furthermore, it was observed that the size of arrested cells did not differ from control cells at the corresponding cell cycle stage (S16 Fig).

To further investigate the effects of KKT2 inhibition on the *Leishmania* cell cycle, we employed fluorescence microscopy to monitor cell cycle progression by staining for β-tubulin (a mitotic spindle component) and DNA. This approach enabled us to classify *Leishmania* promastigotes into eight distinct cell cycle stages based on nuclear and kinetoplast configuration, flagellum number, the presence or absence of a mitotic spindle, β-tubulin distribution, and cytokinesis features (Fig 7). Using this classification system, treatment of parasites expressing the KKT2 AS variant with 5 μM 1NM-PP1 for 6 hours led to an accumulation of cells exhibiting one nucleus, one kinetoplast, one flagellum, and β-tubulin concentrated in the

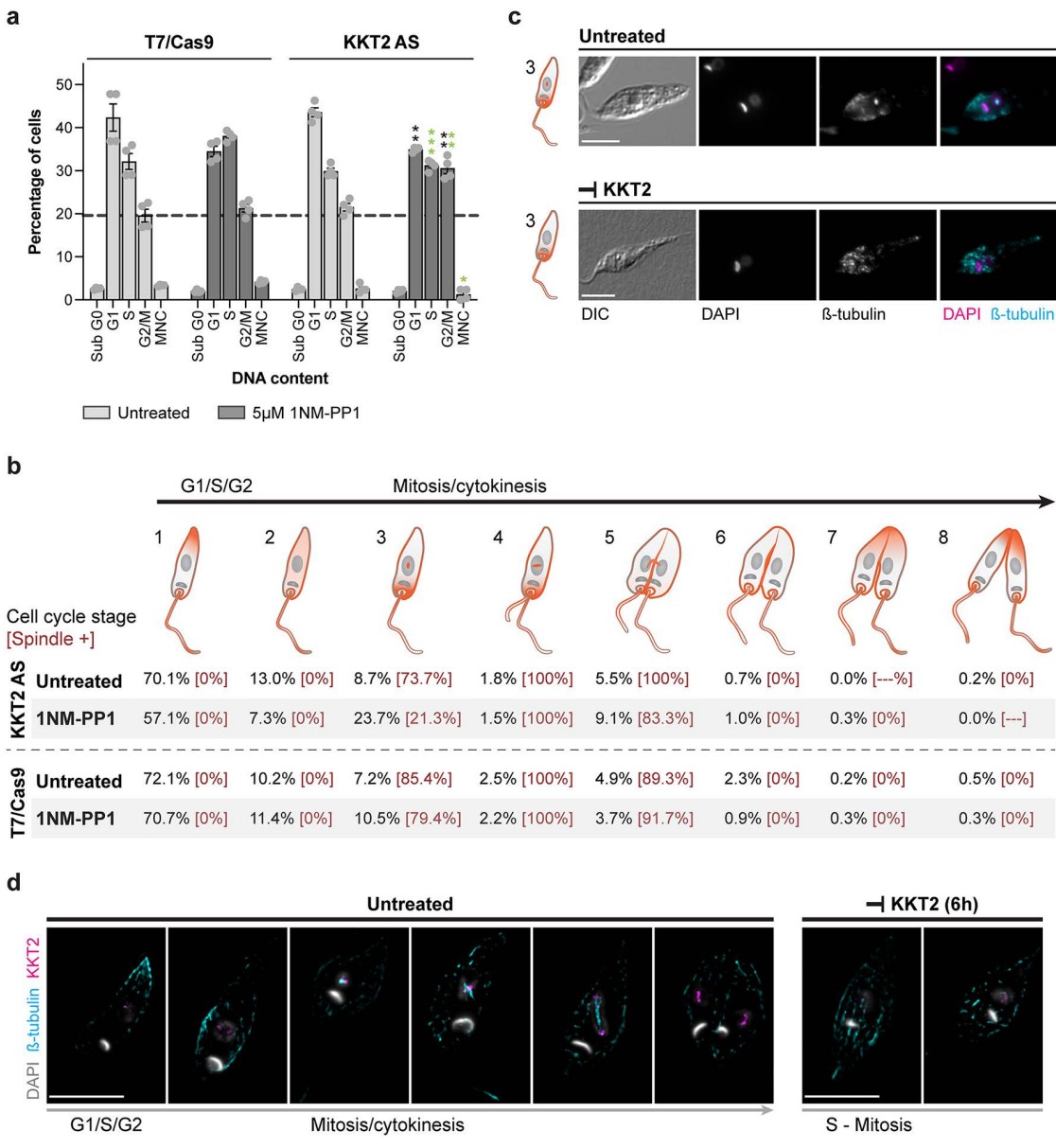

**Fig 6. KKT2 kinase activity is required for cell cycle progression through synthesis and mitotic phases in *Leishmania*. (a)** Effect of KKT2 inhibition on *Leishmania* cell cycle. Bar graph showing the percentage of cells in each cell cycle phase following 6 hours treatment with 5 µM 1NM-PP1. The parental T7/Cas9 line and untreated parasites expressing KKT2$^{M146G}$ cultured under the same conditions were used as controls. Cell cycle phase quantification was performed using the Watson model algorithm in FlowJo v10.10.0. For statistical significance, p-values were calculated using two-tailed Student's t-tests, comparing (i) treated *versus* untreated analog-sensitive populations (black), and (ii) wild-type versus analog-sensitive populations under treated conditions (green) (* p < 0.05; **p < 0.01; *** p < 0.001). Data are mean ± SEM of four biological replicates. G, gap phase; S, synthesis phase; M, mitosis phase; MNC, multinucleated cells; Sub-G0, cells with less DNA than typical G1-phase cells, indicative of DNA degradation. **(b)** Schematic representation of *L. mexicana* cell cycle stages based on nuclear and kinetoplast configurations (grey), flagellum number, β-tubulin (orange) distribution, as well as features of cytokinesis. The percentage of cells in each stage, following 6 hours treatment with 5 µM 1NM-PP1, are given below with the proportion exhibiting mitotic spindles in square brackets (KKT2 AS, n = 396 and 438 cells for treated and untreated conditions, respectively; T7Cas9, n = 324 and 567 cells for treated and untreated conditions, respectively). **(c)** Fluorescence microscopy of promastigote cells treated with 5 µM 1NM-PP1 for 6 hours, stained with the KMX-1 antibody to detect β-tubulin, and counterstained with DAPI to visualize DNA. Representative fluorescence micrographs show cells in stage 3 of the cell cycle (according to panel "b") under treated and untreated conditions. Scale bars, 5 µm. DIC, the Nomarsky differential interference contrast. **(d)** Spatial and temporal distribution of *L. mexicana* KKT2 throughout the cell cycle. High-resolution fluorescence microscopy was performed on KKT2_AS::mNG::3xMyc promastigotes cultured with or without 10 µM 1NM-PP1 for 6 hours. Cells were stained with KMX-1 antibody to

detect β-tubulin, anti-Myc antibody to visualize KKT2, and counterstained with DAPI for DNA visualization. Representative fluorescence micrographs show the localization of KKT2 and β-tubulin across different *Leishmania* cell cycle stages. The colours used for each marker are indicated to the left of the images. Scale bars, 5 μm. Additional individual and merged channel images, along with the untagged T7/Cas9 control line, are presented in S18 Fig.

| Stage | Schematic representation [a] | Description |
|---|---|---|
| 1 | | One nucleus, one kinetoplast, one flagellum; no mitotic spindle; β-tubulin concentrated in the posterior region of the cell. |
| 2 | | One nucleus, one kinetoplast, one flagellum; no visible mitotic spindle; β-tubulin more evenly distributed throughout the cell body. |
| 3 | | One nucleus, one kinetoplast, one flagellum; β-tubulin concentrated in the anterior region; presence or absence of a short mitotic spindle in the centre of the nucleus. |
| 4 | | Nuclear and/or kinetoplast segregation in progress, two flagella (with the newly formed flagellum typically short), β-tubulin concentrated in the anterior region; presence of the mitotic spindle in the nucleus. |
| 5 | | Nuclei and kinetoplasts segregated; two flagella; presence of an elongated mitotic spindle; β-tubulin distributed along the longitudinal axis of the cell; no cleavage furrow. |
| 6 | | Two nuclei, two kinetoplasts, two flagella; absence of the mitotic spindle; β-tubulin aligned along the longitudinal axis; early formation of the cleavage furrow. |
| 7 | | Two nuclei, two kinetoplasts, two flagella; no mitotic spindle; β-tubulin aligned along the longitudinal axis; cleavage furrow extending halfway through the cell body. |
| 8 | | Two nuclei, two kinetoplasts, two flagella; no mitotic spindle; β-tubulin concentrated in th posterior region; near completion of cytokinesis, with the two daughter cells connected only at the posterior tip. |

**Fig 7. Definition of cell cycle stages.** [a] The two DNA-containing organelles – the nucleus and the kinetoplast – are shown in grey. The kinetoplast, the smaller of the two, is located near the base of the flagellum. β-tubulin distribution is depicted in orange.

anterior region of the cell – features characteristic of stage 3 (23.7% compared to 8.7% in the untreated control). However, unlike the untreated control in which most stage 3 cells (73.7%) exhibited a visible mitotic spindle, only 21.3% of stage 3 cells under KKT2 inhibition displayed spindle formation, indicating that KKT2 activity is required either before or during mitotic spindle assembly (Fig 6b–6c). In addition, a small increase in the proportion of cells in stage 5 was observed following KKT2 inhibition (9.1% compared to 5.5% in the untreated control), suggesting that KKT2 activity also plays a role in anaphase progression (Fig 6b).

To investigate the spatial and temporal distribution of KKT2 throughout the cell cycle and under conditions of kinase activity inhibition, we generated a tagged parasite line in which the C-terminus of both alleles of the KKT2 AS gene was endogenously fused to an mNeonGreen (mNG)::3xMyc epitope (KKT2_AS::mNG::3xMyc; S17 Fig). Subsequently,

parasites were cultured in the presence or absence of 10 μM 1NM-PP1 for 6 hours, and KKT2 localization was examined using high-resolution fluorescence microscopy alongside β-tubulin and DNA staining to define cell cycle stages. When the KKT2 kinase is active in *Leishmania* parasites, the KKT2 protein exhibits a weak and diffuse signal during the early stages of the cell cycle, prior to mitotic spindle formation. As the cell cycle progresses – from spindle assembly to chromosome segregation – the KKT2 signal intensifies and becomes distinctly localized at the ends of the mitotic spindle. To investigate KKT2 distribution further, we analysed stage 3 cells lacking a mitotic spindle. In these cells, KKT2 appeared dispersed in the nucleus, in contrast to the untreated control at the same stage, where cells exhibited a developing mitotic spindle with KKT2 concentrated near it (Figs 6d and S18).

Taken together, these results indicate that KKT2 kinase activity is required for key events during the coordination of synthesis and mitotic phases of the cell cycle in *Leishmania*. These experiments were conducted in naturally asynchronous cell populations, which likely explains why the requirement for KKT2 activity was observed across multiple stages of the cell cycle.

## Inhibition of CRK9 kinase activity affects multiple stages of the *Leishmania* cell cycle

We have previously reported that inhibition of *L. mexicana* CRK9 kinase activity disrupts trans-splicing, which is predicted to lead to a widespread reduction in protein expression, resulting in broad cellular defects [46]. To investigate the role of CRK9 kinase activity in cell cycle progression, promastigotes expressing the AS variant of CRK9 were treated with 5 μM 1NM-PP1 for 6 hours. After treatment, a significant accumulation of cells in the G1 and G2/M phases was observed, along with a marked reduction in cells in S phase. Furthermore, CRK9 inhibition led to an increase in the proportion of cells in the Sub G0 phase, which is commonly associated with reduced DNA content, suggesting that CRK9 inhibition leads to DNA abnormalities (Figs 8a and S19a). Notably, after 24 hours treatment, the significant accumulation in the Sub G0 phase persisted. The sustained increase in the Sub G0 phase after 24 hours indicates that the effects of CRK9 inhibition accumulate over time, leading to irreversible damage to the parasite. At this time point, an accumulation of multinucleated cells was additionally observed upon CRK9 inhibition. Comparison of treated CRK9 analog-sensitive parasites with treated wild-type controls further indicated a sustained reduction in the proportion of cells in S phase (S19b Fig).

We applied our cross-linking BioID (XL-BioID) proximity labelling strategy to map the CRK9 proximal proteome and uncover potential functional pathways involving CRK9 [7,46]. To this end, we generated 3xMyc::BirA*::CRK9 and the spatial reference 3xMyc::BirA*::BDF7 (Bromodomain Factor 7) cell lines using CRISPR-Cas9. A total of 137 statistically significant CRK9-proximal proteins were identified, including the previously reported CRK9 interactor, cyclin CYC12 [53] (Fig 8b). Gene Ontology (GO) enrichment analysis revealed that the CRK9-proximal proteome is predominantly composed of proteins involved in DNA and RNA processing, consistent with CRK9's role in gene expression regulation. While the exact mechanism by which CRK9 regulates trans-splicing is not fully established, in *T. brucei* CRK9 has been reported to phosphorylate the N-terminal region of the spliceosomal protein SF3B1, and that its kinase activity is essential for spliceosome activation [54]. We observed enrichment of SF3B1 (LmxM.28.2570) among the XL-BioID proximal protein hits (S2 Data), indicating that this dataset can contain potential substrates of CRK9 as well as proteins simply in close proximity. It has also been postulated that CRK9 may act on other substrates that lie upstream of splicing factors in transcription-coupled processes, given that splicing and polyadenylation occur co-transcriptionally in kinetoplastids [30,53]. In addition, CRK9 has been shown to be necessary for phosphorylation of the CTD of RBP1 in *T. brucei*, but this loss of phosphorylation did not impact Pol II transcriptional activity [55]. Indeed, we see enrichment of components of the Polymerase Associated Factor 1 complex (Leo1, CTR9), components that could be consistent with Elongator (Acetyltransferase-like, WD40-like), elongation factors (Spt5, TFIIS), and various components of RNA polymerase complexes (S2 Data). We also find components involved in DNA mismatch repair (MSH2, MSH3), PCNA and DNA polymerase delta, which could reflect a transcription-coupled mismatch repair

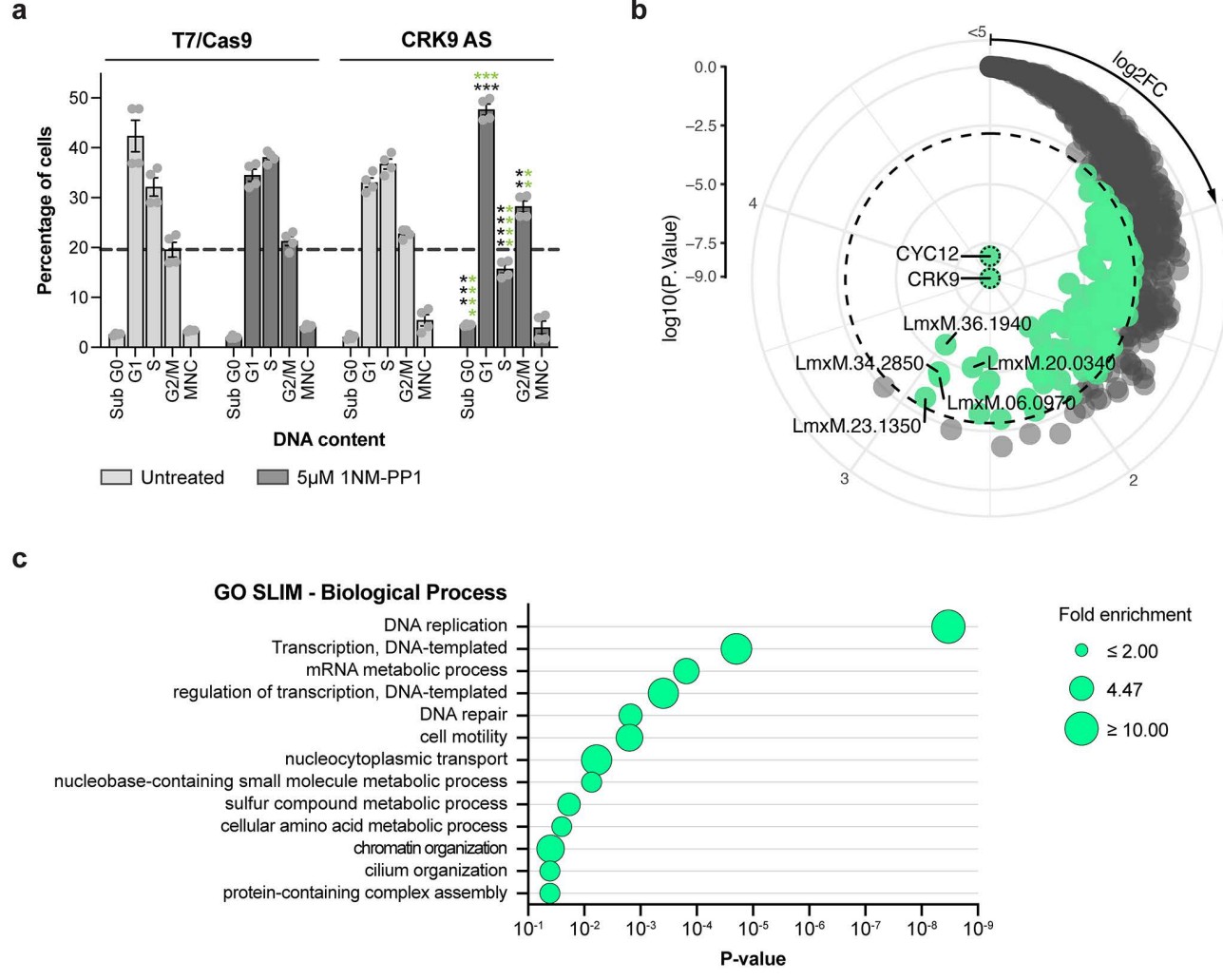

**Fig 8. CRK9 protein kinase activity in the *Leishmania* cell cycle and its proximal proteome. (a)** Effect of CRK9 inhibition on *Leishmania* cell cycle. Bar graph showing the percentage of cells in each cell cycle phase following 6 hours treatment with 5 μM 1NM-PP1. The parental T7/Cas9 line and untreated parasites expressing CRK9$^{M501G}$ cultured under the same conditions were used as controls. Cell cycle phase quantification was performed using the Watson model algorithm in FlowJo v10.10.0. For statistical significance, p-values were calculated using two-tailed Student's t-tests, comparing (i) treated *versus* untreated analog-sensitive populations (black), and (ii) wild-type versus analog-sensitive populations under treated conditions (green) (**$p < 0.01$; ***$p < 0.001$; ****$p < 0.0001$). Data are mean ± SEM of four biological replicates. G, gap phase; S, synthesis phase; M, mitosis phase; MNC, multinucleated cells; Sub-G0, cells with less DNA than typical G1-phase cells, indicative of DNA degradation. **(b)** CRK9-proximal proteins. Proximal proteins were determined with the limma package [56], with multiple testing correction performed using the Benjamini-Hochberg procedure. Radial plot show proteins enriched in CRK9 compared to the spatial reference BDF7. The dashed line indicates a 1% false discovery rate (FDR) threshold used to distinguish proximal (green) from non-proximal (grey) proteins. Log2 fold enrichment increases in a clockwise direction; Log10 p-value increases from the centre outward. The six most enriched proteins (log2FC) are labelled in the plot: LmxM.36.5640, cyclin CYC12; LmxM.36.1940, inosine-guanosine transporter; LmxM.34.2850, RNA polymerase-associated protein LEO1; LmxM.23.1350, acetyltransferase-like protein; LmxM.06.0970, Domain of unknown function (DUF3883); LmxM.20.0340, hypothetical protein. **(c)** Core biological processes of the CRK9-proximal proteome. Bubble plot showing the significantly enriched Gene Ontology (GO) biological processes among CRK9-proximal proteins.

pathway, or be involved in repairing damage caused by R-loops. The analysis also highlighted potential involvement of CRK9 in more distinct biological processes, such as cell motility, cilium organization, and amino acid metabolism (Fig 8c) but these remain to be explored in detail.

## Identification of *Leishmania* protein kinases with targetable cysteines

Having exploited the gatekeeper residue to utilise the AS chemical genetics system in *Leishmania*, we decided to further explore the active site of *Leishmania* protein kinases using a broad-spectrum kinase inhibitor SM1–71. This probe acts by targeting the ATP-binding site and contains an acrylamide moiety that forms covalent adducts with cysteine residues. SM1–71 was originally developed to covalently bind cysteine residues immediately N-terminal to the DFG motif (DFG-1) in human kinases with its biotinylated analog, SM1–71-biotin enabling efficient capture on streptavidin-coated magnetic beads [38,39]. We had previously used a BRET-based target engagement assay to identify cell permeable ATP-competitive inhibitors of *L. mexicana* CLK1/2 and identified WZ8040, an inhibitor with close structural similarity to SM1–71 [40] (S20 Fig). On this basis, we hypothesised that SM1–71 could further facilitate the characterisation of CLK1 as a drug target, particularly given that its AS variants are catalytically inactive. More broadly, we reasoned that this approach could enable systematic identification of reactive cysteine residues across the *Leishmania* kinome, thereby providing a valuable resource to inform the development of covalent inhibitors. So, we initially performed a label-free MS/MS analysis of proteins enriched from promastigote cell lysates incubated with either DMSO or SM1–71-biotin (Fig 9a), with the expectation of identifying CLK1 as one of the targets. This analysis identified 868 *Leishmania* proteins, which were consolidated into 802 unique entries based on sequence similarity for relative quantification. Among these, 131 proteins (121 entries) were significantly enriched in the SM1–71-biotin condition. Notably, 29 of the enriched proteins (28 entries) were annotated as protein kinases (including CLK1/CLK2), and 38 entries were identified as other nucleotide-binding proteins. The enrichment of non-kinase proteins is likely attributed to the probe capturing hyper-reactive cysteines present throughout the proteome [57,58]. Structural predictions using AlphaFold 3 revealed that 14 of these kinases possess cysteine residues in proximity to the DFG motif – 10 possess a cysteine at the canonical DFG-1 position, and 4 have cysteines located within seven residues of the motif. Additionally, 15 kinases were found to contain cysteine residues within the P-loop region (S3 Data and S21 Fig).

To strengthen these findings, we applied a competitive chemoproteomic approach in which promastigote lysates were pre-treated with unlabelled SM1–71 (lacking biotin but including the acrylamide warhead) prior to incubation with SM1–71-biotin. This approach enabled the identification of more stable and specific probe-protein interactions. Through this strategy, 12 proteins were significantly enriched by SM1–71-biotin, including 11 protein kinases (S3 Data and S21 Fig). Of these, 10 protein kinases overlapped with those identified in the initial enrichment assay. These consistently enriched kinases were classified as high-confidence candidates containing targetable cysteine residues amenable to covalent inhibition.

The sequence and structural prediction revealed the presence of a cysteine at the DFG-1 position in CLK1/CLK2 (C292 in CLK1, which differs from the cysteine C219 targeted by the covalent inhibitor AB1 [7]), MPK4 (C173), MPK5 (C193), MPK7 (C272), and the protein kinase LmxM.32.1710 (C172). This DFG-1 cysteine should be the major site of covalent interaction with the SM1–71 inhibitor, as the probe was specifically designed to target this region of the ATP-binding pocket [39]. To identify the SM1–71 binding site(s) on CLK1, rCLK1 was incubated with SM1–71 and analysed by LC-MS/MS. In total, nine cysteine residues were detected as modified by SM1–71 upon treatment: C159, C210, C219, C287, C292, C297, C298, C392 and C417 (Fig 9b). Comparison of peptide intensities between SM1–71-treated and untreated samples revealed that, contrary to expectation, SM1–71, predominantly targets C219, located in the hinge region of the ATP-binding pocket, and C392, which lies outside the ATP-binding site. These residues exhibit the highest intensities of SM1–71 modification, accompanied by the most consistent reduction in alkylation modification (Fig 9c–9d). These findings validate the SM1–71-biotin pull-down results and demonstrate that CLK1 contains multiple reactive cysteine residues that may be exploited for covalent inhibitor development.

Additionally, we observed enrichment of protein kinases that lack the DFG-1 cysteine but contain cysteines either in the P-loop region or in proximity to the DFG motif, sites previously shown to be reactive with the SM1–71 probe [38]. Among them are AEK1, which harbours cysteine C63 in the P-loop and C185 at the DFG-4 position; and the NEK family members

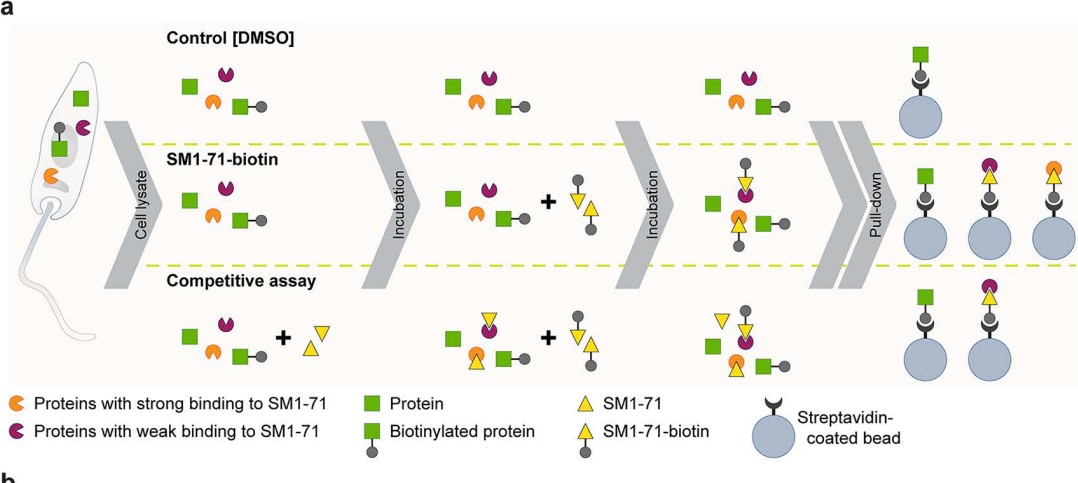

**a**

Control [DMSO]

SM1-71-biotin

Competitive assay

Cell lysate — Incubation — Incubation — Pull-down

🟠 Proteins with strong binding to SM1-71  🟩 Protein  🔺 SM1-71

🟣 Proteins with weak binding to SM1-71  🟩● Biotinylated protein  🔺● SM1-71-biotin

🔵 Streptavidin-coated bead

**b**

| Cys | Peptide | Cys | Peptide |
|-----|---------|-----|---------|
| 159 | KEY**c**AVK; RKEY**c**AVK | 292 | I**c**DLGGCCDER; I**c**DLGGC**c**DER; I**c**DLGG**cc**DER; VRI**c**DLGGCCDER |
| 210 | YFQNDSGHM**c**IVMPK | 297 | ICDLGG**c**CDER; ICDLGG**cc**DER; I**c**DLGG**c**CDER; VRICDLGG**c**CDER |
| 219 | YGP**c**LLDWIMK | 298 | ICDLGGC**c**DER; ICDLGG**cc**DER; I**c**DLGGC**c**DER |
| 287 | HLPPDP**c**R | 392 | LLYNSAGQLRP**c**TDPK |
| | | 417 | DVIRDDLL**c**DLIYGLLHYDR; DVIRDDLL**c**DLIYGLLHYDRQK; TVRDVIRDDLL**c**DLIYGLLHYDR |

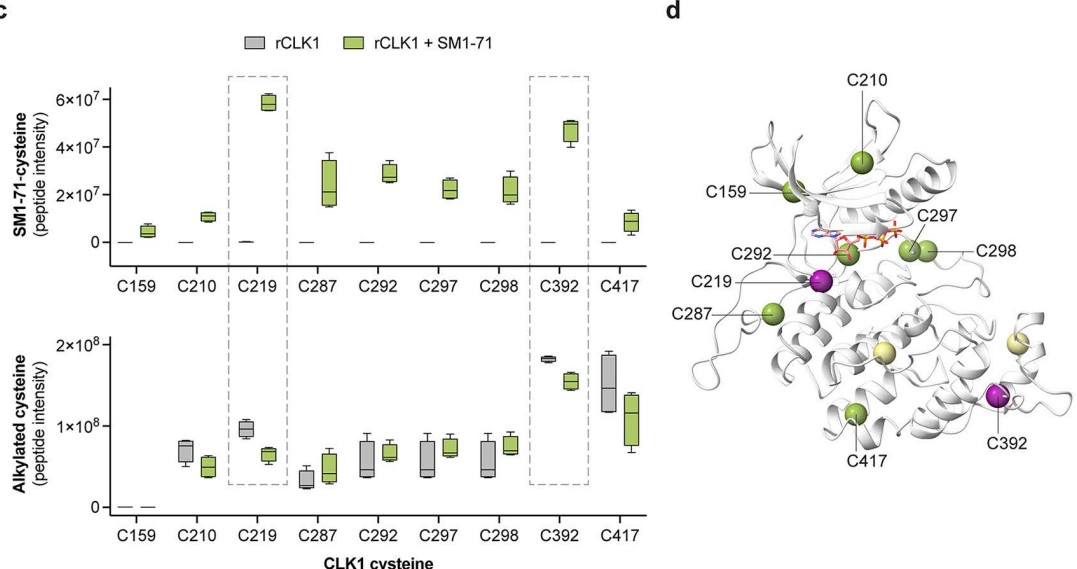

**c** □ rCLK1  □ rCLK1 + SM1-71

**d**

**Fig 9. Targetable cysteine residues in *L. mexicana*. (a)** Schematic overview of the pull-down strategies used to identify protein kinases harbouring reactive cysteine residues. **(b)** Identification of SM1−71-modified cysteines in rCLK1. Purified rCLK1 was incubated with SM1−71, and covalently modified peptides were identified by LC-MS/MS following reduction with DTT, alkylation with iodoacetamide, and in-gel digestion with trypsin. Peptides containing SM1−71-cysteine adducts are listed in the table. The modified cysteine is indicated in magenta and in lower case **(c)** C219 and C392 are the predominant sites of SM1−71 modification in rCLK1. Intensities of SM1−71-modified peptides are shown in the top panel, whereas intensities of iodoacetamide-modified peptides are shown in the bottom panel. Box plots show the interquartile range (25th - 75th percentiles), with the mean indicated by the line within the box and whiskers extending to the minimum and maximum values. **(d)** Location of reactive cysteine residues in *L. mexicana* CLK1. The CLK1 structural model was generated using AlphaFold 3 [43] and visualised in ChimeraX v1.9. The ATP ligand is shown as a stick representation with heteroatom colouring. Cysteine Cα atoms are depicted as spheres: green indicates residues identified by LC-MS/MS as forming covalent adducts with SM1−71, and purple highlights the cysteines most consistently modified.

LmxM.36.1520 (C52, C66, C98) and LmxM.36.1530 (C103), which contain cysteines in the P-loop region. Curiously, protein kinases STK36 and LmxM.19.0150 contain cysteines only in the C-lobe of the kinase domain, distant from the active site, suggesting the presence of available-reactive cysteines in these proteins. Five of these protein kinases with targetable cysteine residues in the active site were predicted to be essential for promastigote and/or amastigote survival: the kinetochore protein CLK1/CLK2, the mitogen-activated protein kinases MPK4, MPK5, and MPK7, and the AGC essential kinase AEK1 (Table 2).

## Discussion

Owing to their well-established druggability, protein kinases are among the most intensively investigated targets in pharmaceutical research, with over 80 protein kinase inhibitors approved by the U.S. Food and Drug Administration (FDA) for the treatment of various cancers and other diseases [12,59]. Recently we genetically validated over 40 protein kinases from the *Leishmania* kinome as they were found to be required for parasite survival [14]. Kinase druggability has also been demonstrated in *Leishmania*, where a pyrazolopyrimidine compound was shown to selectively inhibit CRK12, a member of the CMGC kinase family [5], and the amidobenzimidazole compound AB1 was found to specifically target the kinetochore-associated kinases CLK1/CLK2 [7].

Our kinome-wide bioinformatics analysis of *L. mexicana* revealed the absence of protein kinases with naturally occurring glycine or alanine residues at the gatekeeper position. This finding stands in contrast to several apicomplexan parasites, including *Toxoplasma gondii* [60], *Cryptosporidium parvum* [61], and *Neospora caninum* [62], where glycine residues are found at the gatekeeper position in calcium-dependent protein kinase 1 (CDPK1), thereby enabling selective

**Table 2.** *L. mexicana* protein kinases with targetable cysteines.

| Accession | Kinase Group/Family | Kinase Name | Cysteine position in the kinase ATP-binding site [a] | | | | |
|---|---|---|---|---|---|---|---|
| | | | P-Loop | C-Helix | Hinge | DFG | Outside [b] |
| LmxM.09.0400 * | CMGC/CLK | CLK1/ KKT10 | C159[P4] C210[P6] | | **C219** | **C292[DLG-1]** C297[DLG+2] C298[DLG+3] | |
| LmxM.13.0440 | Other/ULK | STK36 | | | | | C192[C Lobe] C218[C Lobe] |
| LmxM.13.1640 * | CMGC/MAPK | MPK7 | | | | C267[DFG-6] **C272[DFG-1]** | |
| LmxM.19.0150 | STE | | | | | | C105 |
| LmxM.19.1440 * | CMGC/MAPK | MPK4 | **C41[P2]** C45[P3] | | | C168[DFG-6] **C173[DFG-1]** | |
| LmxM.25.2340 * | AGC | AEK1 | C63[P3] | | | C185[DFG-4] | |
| LmxM.29.2910 * | CMGC/MAPK | MPK5 | C58[P3] | C100[+1] C102[+2] | | **C193[DFG-1]** | |
| LmxM.32.1710 | CAMK/CAMKL | | | | | **C172[DFG-1]** | |
| LmxM.36.1520 | NEK | | C52[P3] C66[P4] C98[P5] | C86 C90[+2] | | | C179 C208 C222 |
| LmxM.36.1530 | NEK | | C103[P5] | C90 C91 C95[+2] | | | C184 **C213** |

* *L. mexicana* protein kinase required for promastigote and/or amastigote survival (S1 Data). [a] The positions of cysteine residues surrounding the ATP-binding site were identified using AlphaFold 3 structural prediction and visualised with ChimeraX v1.9. The ribbon-structure representation of cysteine locations is shown in S21 Fig. Cysteines located within 6 Å of the ATP ligand are highlighted in bold. [b] Outside, cysteine located outside the ATP-binding site.

inhibition by bumped kinase inhibitors and supporting their development in a targeted therapeutic approach. Selectivity in these models can be validated by substituting the gatekeeper residue with a bulkier amino acid, thereby blocking inhibitor access to the ATP-binding pocket, as demonstrated for *Plasmodium falciparum* cGMP-dependent protein kinase (PKG) [63]. Notably, *Leishmania* does not contain this class of calcium-dependent protein kinases. Motivated by the absence of naturally occurring AS-compatible kinases in *L. mexicana*, we adopted a chemical-genetic strategy to conditionally interrogate kinase function. The smallest gatekeeper residue identified in our kinome analysis was a serine in the CMGC family kinase LmxM.36.4250, which is required for amastigote survival in an *in vivo* mouse model, but not in promastigotes or *in vitro* macrophage-internalized amastigotes. In addition, we identified 15 kinases with a threonine at the gatekeeper position (LmxM.15.1550, LmxM.05.0130, LmxM.10.0200, LmxM.10.0490, LmxM.13.1640, LmxM.14.1070, LmxM.21.1650, LmxM.29.0370, LmxM.29.2910, LmxM.02.0360, LmxM.21.0823, LmxM.28.0620, LmxM.21.0853, LmxM.31.1810, and LmxM.07.0880). Among these, LmxM.13.1640, LmxM.21.1650, LmxM.29.0370, and LmxM.29.2910 were classified as required for amastigote survival *in vivo* model. This observation is relevant given prior evidence that naturally occurring serine and threonine gatekeeper residues can confer sensitivity to BKIs, as shown for MAPKL-1 (mitogen-activated protein kinase like 1) in *T. gondii* [64], and CDPK1 and CDPK4 in *Plasmodium spp* [65,66]. These findings have important implications for the application of AS kinase technology in *Leishmania*. While *L. mexicana* represents a suitable model for in-cell target validation using AS kinase approaches in promastigote and *in vitro* macrophage-internalized amastigotes, the presence of the natural serine and threonine AS kinases may confound interpretation in *in vivo* models. Furthermore, although none of these kinases were classified as required in promastigotes or *in vitro* amastigotes, the possibility remains that simultaneous inhibition of multiple serine- or threonine-gatekeeper kinases contributes to the $IC_{50}$ values measured in wild-type parasites (11.42 μM in promastigotes and 19.63 μM in *in vitro* amastigote infections for 1NM-PP1).

We selected four kinetochore-associated kinases, CLK1/CLK2, KKT2 and KKT3, along with the cyclin-dependent kinase CRK9, for chemical-genetic validation of their kinase activity requirements in *Leishmania* using the inducible AS kinase system. We were unable to generate CLK1/CLK2 AS mutants, probably because the mutation is not tolerated by the kinase. Previous studies have reported that substitution of the gatekeeper residue can markedly reduce kinase activity, an effect that can be rescued by introducing a second-site suppressor mutation. For example, in human G protein-coupled receptor kinase 2 (GRK2), like *Leishmania* CLK1, contains a leucine rather than a phenylalanine in the DFG motif (DLG), and restoration of analog-sensitive kinase activity to approximately 82% of wild-type levels was achieved through rescue of the DFG domain [67]. In this context, introducing such a compensatory mutation could be explored to obtain functional CLK1/CLK2 AS mutants. Using the AS approach, we demonstrated that the kinase activities of KKT2 and CRK9 are essential for parasite survival in both the promastigote stage and, critically, in the clinically relevant intracellular amastigote stage. These findings position KKT2 and CRK9 as promising therapeutic targets for the development of anti-leishmanial kinase inhibitors. Notably, KKT2 lacks orthologs in the human kinome [47], indicating a greater potential for achieving selectivity in drug development. In contrast, CRK9 has sequence similarity to many human CDKs, with CDK2 showing the highest sequence identity (37%; S22 Fig). Supporting our findings, CRK9 kinase activity has previously been validated as essential for parasite survival in a murine model of *T. brucei* infection [53]. Importantly, we also showed that the in-cell AS kinase strategy is compatible with the intra-macrophage amastigote stage, enabling conditional chemical modulation of kinase activity within an intracellular environment. This represents a significant advance, as direct genetic manipulation of intracellular amastigotes remains technically challenging, limiting functional assessment of kinase essentiality for parasite survival.

In contrast, AS-mediated inhibition of KKT3 did not result in any observable growth defect in *Leishmania*. The KKT3 protein in *T. brucei* contains, in addition to its kinase domain, a centromere localization domain and a divergent polo box domain, the latter of which has been shown to be essential for proliferation in the procyclic stage [52]. Sequence alignment between *L. mexicana* and *T. brucei* KKT3 indicates that these domains are conserved across trypanosomatids, suggesting a conserved functional architecture. Nevertheless, given that our attempts to generate a catalytically inactive

KKT3 mutant were unsuccessful, it is likely that KKT3 kinase activity is required for *Leishmania* parasite survival and that the gatekeeper substitution alone did not sensitize the mutant to inhibition by the bumped kinase inhibitors. Further investigation will therefore be required to validate KKT3 as a potential therapeutic target, particularly in the clinically relevant amastigote stage. In this context, introducing a secondary mutation alongside the gatekeeper substitution, which has been reported to further tailor the ATP-binding pocket and confer sensitivity to BKIs, could be explored [68]. Interestingly, in *T. brucei*, KKT3 kinase activity has been reported to be dispensable for proliferation of procyclic cells [52], whereas RNAi-mediated depletion of the full-length protein indicated that KKT3 is required for parasite survival [47,69].

We leveraged our inducible AS kinase system to investigate the functional roles of KKT2 and CRK9 in *Leishmania*, aiming to gain mechanistic insights into the biology of these proteins. KKT2 is a non-canonical protein kinase and a core component of the kinetoplastid kinetochore [47] – a specialized macromolecular complex that, together with centromeres and spindle microtubules, drives chromosome segregation in eukaryotes [70]. In *T. brucei*, RNAi-mediated depletion of KKT2 has been shown to impair proliferation in both procyclic and bloodstream forms, and to disrupt the localization of the kinetochore protein KKT14 [69,71]. Subsequent studies demonstrated that KKT2 kinase activity is specifically required for proliferation in the bloodstream stage [41] and for accurate chromosome segregation in the procyclic form [52]. However, KKT2 kinase activity was found to be dispensable for the localization of KKT2 itself, as well as for the proper localization of the kinetochore proteins KKT1, KKT9, and KKT14 [41,52]. KKT2 is a multidomain protein and its unique centromere localization domain in the middle, and its divergent polo-box domain at the C-terminus have been shown to be sufficient for its correct localization [71,72]. Our findings revealed that KKT2 kinase activity is essential for mitotic progression in *L. mexicana*, as evidenced by an increased proportion of cells arrested at an early mitotic stage (stage 3) lacking a detectable mitotic spindle upon inhibition of AS mutant. This observation suggests that KKT2 activity may be required for mitotic spindle assembly. However, given that KKT2 function has primarily been linked to kinetochore assembly [41,47,71], and that we did not directly assess the impact of KKT2 inhibition on kinetochore assembly, we cannot exclude the possibility that the spindle defects observed are secondary to impaired kinetochore function resulting from loss of KKT2 activity. In addition, we demonstrate that KKT2 activity is required for proper coordination of S-phase progression, providing new insights into the regulation of cell cycle transitions in kinetoplastids. We found that KKT2 is constitutively expressed throughout the *Leishmania* cell cycle and displays a punctate nuclear signal. Prior to mitosis, KKT2 exhibits a more diffuse nuclear distribution, whereas during mitosis it becomes concentrated at the ends of the mitotic spindle, suggesting a spatially dynamic role linked to mitotic progression.

To further dissect essential regulatory nodes in the *Leishmania* cell cycle, we investigated CRK9, a member of the cyclin-dependent kinases, which play central roles in regulating cell cycle progression and gene expression in eukaryotes [73]. Although trypanosomatid cyclin-dependent kinases retain a conserved domain architecture, their primary sequences are divergent, making it difficult to unambiguously identify their counterparts in human and yeast [13]. Like other cyclin-dependent kinases, CRK9 functions in complex with a cyclin (CYC) partner, CYC12, an L-type cyclin, and together with CRK9-associated protein (CRK9AP), forms an unusual tripartite complex [53]. Although CRK9AP was not detected in our proximity labelling dataset, CYC12 was the most enriched CRK9-proximal protein. In *T. brucei*, CRK9 has been implicated in the regulation of pre-mRNA processing [55], an effect later shown to depend directly on its kinase activity [30]. Consistent with this role, our previous work demonstrated that inhibition of CRK9 kinase activity in *Leishmania* leads to accumulation of unspliced pre-mRNAs, indicating a conserved role in RNA processing among kinetoplastids [46]. In the present study, we found significant enrichment of CRK9-proximal proteins involved in DNA and RNA processing pathways. Furthermore, we show that AS-mediated inhibition of CRK9 in *Leishmania* results in broad disruption of cell cycle progression, marked by accumulation of cells in Sub-G0, G1, and G2/M phases, and a corresponding reduction in S-phase cells. This pleiotropic phenotype is consistent with the expected consequences of inhibition of trans-splicing, which disrupts global mRNA maturation and protein synthesis, ultimately leading to broad cellular dysfunction. Notably, the increase in the Sub-G0 population, often associated with DNA fragmentation, persisted after 24 hours of treatment, suggesting that

CRK9 inhibition may lead to DNA content dysregulation. Consistent with our findings, RNAi-mediated depletion of CRK9 in *T. brucei* has been shown to impair mitosis and cytokinesis in the procyclic form, although no significant impact on proliferation was observed in the bloodstream stage [74].

In addition, our kinome-wide bioinformatic analysis of gatekeeper residues also identified alternative opportunities for chemical-genetic intervention. Notably, we found that the protein kinase LmxM.36.0910, a member of the STE kinase family, harbours a cysteine at the gatekeeper position – a feature that renders it susceptible to covalent inhibition by electrophilic compounds specifically designed to target gatekeeper cysteines [75]. This discovery provides a potential route to functionally interrogate LmxM.36.0910 using covalent chemical tools, however, functional genetics analyses revealed that this kinase is non-essential across all life cycle stages of *Leishmania* parasite, indicating limited therapeutic relevance despite its biochemical tractability.

In the context of covalent chemical tools, our chemoproteomic profiling using the electrophilic probe SM1–71 identified five essential kinases, CLK1/CLK2, MPK4, MPK5, MPK7, and AEK1, that harbour reactive cysteine residues amenable to covalent inhibition. Independent genetic and functional studies have pinpointed the essentiality of these kinases. In *L. mexicana*, MPK4 has been shown to be required for amastigote survival, as deletion of the endogenous gene was only achievable in the presence of a functional episomal copy, which was stably maintained during prolonged infection in mice even in the absence of selective pressure [76]. In addition, although *L. mexicana* MPK5 null mutants were successfully generated by classical homologous recombination and persisted at the inoculation site in BALB/c mice, loss of MPK5 was associated with reduced parasite fitness, including impaired amastigote proliferation and reduced lesion formation [77]. Interestingly, *in vivo* studies demonstrated a marked reduction of the virulence in BALB/c mice in both MPK7 null and MPK7 overexpressing *L. mexicana* mutants [78]. Consistent with these findings, overexpression of MPK7 in *L. major* led to delayed lesion formation and reduced amastigote proliferation [79]. Finally, AEK1 has been highlighted as a potential drug target in *T. brucei* [32] and *T. cruzi* [33].

Covalent kinase inhibitors represent an emerging class of therapeutics, with eleven FDA-approved compounds employing irreversible targeting of cysteine residues through Michael addition chemistry [59,80]. In line with this strategy, our previous work demonstrated that AB1 and WZ8040 covalently bind to a hinge-region cysteine in CLK1 of trypanosomatids [6,7,40]. Altogether, these findings support the further validation of MPK4, MPK5, MPK7 and AEK1, using our analog-sensitive chemical genetics platform, as promising candidates for the development of covalent kinase inhibitors in *Leishmania*.

To enable functional interrogation of protein kinase activity in *Leishmania*, in this study, we successfully adapted a CRISPR-Cas9-based genome editing strategy to introduce single-codon substitutions directly into endogenous *loci* without the use of drug selection markers. Our marker-free approach employed 120-nt single-stranded oligonucleotide donor DNA containing short homology arms, the desired target mutation, and silent shield mutations to prevent further Cas9-mediated cleavage. Two sgRNAs flanking the target codon were used in our strategy. Using this system, we achieved an average editing efficiency of 11.7% for the successful substitution of the codon encoding the gatekeeper residue in both alleles of *KKT2*, *KKT3*, and *CRK9*, demonstrating that exhaustive screening was not required to isolate correctly edited clones for these genes. Comparable approaches have been reported in other kinetoplastids. In *T. brucei*, a similar strategy was used to introduce point mutations into the aquaglyceroporin gene (AQP2) using a 48-nt single-stranded oligonucleotide repair template and a sgRNA expressed from a vector, yielding an editing efficiency of approximately 8% [81]. In *T. cruzi,* CRISPR-Cas9-mediated editing using single-stranded oligonucleotide DNA donors has also been used to introduce premature stop codons, although this system relied on the delivery of recombinant Cas9 ribonucleoprotein complexes rather than endogenous Cas9 expression [82]. In *Leishmania*, marker-free genome editing was first demonstrated in *L. donovani*, where a single-stranded oligonucleotide was used to introduce a stop codon into the miltefosine transporter gene, which conferred resistance to miltefosine [83]. Unlike our strategy, in that system the sgRNA was expressed from an episomal vector and clones carrying the desired mutation were enriched by miltefosine selection. A subsequent

adaptation of this approach employed a co-targeting strategy, in which the miltefosine transporter gene was simultaneously targeted to allow for selection-based enrichment of edited cells. This co-selection significantly improved editing efficiency and enabled the deletion of multicopy gene families in *L. donovani* [84]. Furthermore, a marker-free CRISPR-Cas9 editing strategy was described in *L. major* to introduce a single amino acid substitution in the calcium-dependent kinase SCAMK, with both the sgRNA and DNA donor delivered via an episomal expression vector [85]. Unlike these systems, our approach is fully compatible with the high-throughput CRISPR-Cas9 toolkit developed by Beneke et al. [44], which enables streamlined genome editing without the need for custom vector construction. This compatibility, combined with the absence of selection markers, makes our strategy well-suited for scalable functional genomic studies in *Leishmania*.

Altogether, this study establishes a powerful framework for dissecting protein kinase function in *Leishmania*, integrating chemical-genetic tools with CRISPR-mediated precise genome editing to enable in-cell interrogation of targeted kinases. Through this approach, we identify KKT2 and CRK9 as promising therapeutic targets for antileishmanial drug development. Notably, the evolutionary conservation of these protein kinases across clinically relevant trypanosomatid species raises the prospect of developing a single therapeutic agent with efficacy against multiple kinetoplastid diseases, including leishmaniasis, Chagas disease, and African trypanosomiasis.

## Materials and methods

### Ethics statement

All experiments were conducted according to the Animals (Scientific Procedures) Act of 1986, United Kingdom, and had approval from the University of York Animal Welfare and Ethical Review Body (AWERB) committee.

### Cell culture

*Leishmania mexicana* promastigotes (strain MNYC/BZ/62/M379) were cultured at 25°C in HOMEM medium (modified Eagle's medium; Gibco, ThermoFisher Scientific) supplemented with 10% heat-inactivated fetal calf serum (hi-FCS) (Gibco, ThermoFisher Scientific) and 100 U penicillin – 100 µg mL$^{-1}$ streptomycin (Sigma-Aldrich), pH 7.2. Where applicable, selective antibiotics were added at the following concentrations: Hygromycin B (InvivoGen, ant-hg) at 50 µg mL$^{-1}$; Nourseothricin (Jena Bioscience, AB-101) at 50 µg mL$^{-1}$; Blasticidin (InvivoGen, ant-bl) at 10 µg mL$^{-1}$; Puromycin (InvivoGen, ant-pr) at 30 µg mL$^{-1}$. Bone marrow-derived macrophages (BMDM) isolated from BALB/c mice were differentiated in DMEM (Dulbecco's Modified Eagle Medium) medium supplemented with 10% hi-FCS and macrophage colony-stimulating factor secreted by L929 cells [86]. BMDM was maintained in culture in DMEM medium supplemented with 10% hi-FBS and 10 mM L-glutamine (Gibco, ThermoFisher Scientific) at 37ºC, in an atmosphere of 5% $CO_2$.

### Growth curve

*L. mexicana* promastigotes were inoculated at a density of $4 \times 10^4$ parasites mL$^{-1}$ in HOMEM medium supplemented with 10% hi-FCS. The cumulative cell growth was monitored daily by manual counting using a Neubauer hemocytometer. Growth rate calculations were performed using data from the logarithmic phase of the growth curve (0 – 96 h).

### Identification of kinase gatekeeper residues in the *Leishmania* kinome

The kinase domains of the 216 predicted protein kinases in the *L. mexicana* genome were identified using InterPro domain analysis [87]. The gatekeeper residue was identified by multiple sequence alignment of each kinase family. Where the alignment was ambiguous, or an unusual residue was predicted, the gatekeeper was confirmed by 3D structure comparison with a kinase of known structure and a well-defined gatekeeper (e.g., PDB code 3F3W). When a suitable structure model (e.g., of its paralog from *L. infantum*) for the kinase in question was not available in the AlphaFold database, a model was generated using AlphaFold 3 [43]. Structures were aligned using secondary structure matching

[88] and the superposition was checked by confirming the overlap of conserved motifs around the active site Lys and the conserved Asp residues in the sequence motifs HRD and DFG. The gatekeeper residue was identified as being at the end of a β-strand immediately before the hinge region that separate the N- and C-lobes of the protein kinase. The predicted models were visualized in ChimeraX v1.9.

### *L. mexicana* genome editing by CRISPR-Cas9

To achieve precise genome editing, we utilized the CRISPR-Cas9 system in *L. mexicana* strain MNYC/BZ/62/M379, which constitutively expresses Cas9 and T7 RNA polymerase, as previously described [44,89]. To introduce single-codon mutations without the use of drug selection markers, two single-guide RNAs (sgRNAs) targeting sequences flanking the codon of interest was employed. The sgRNAs were designed using the Eukaryotic Pathogen CRISPR guide RNA/DNA Design Tool (http://grna.ctegd.uga.edu) with default parameters (SpCas9: 20 nt gRNA, NGG PAM on 3'end). Primers for sgRNA synthesis were manually designed to align with the high-throughput system outlined by Beneke et al. [44]. The sgRNA templates were generated via PCR using a forward oligonucleotide encoding the T7 promoter sequence (5'-GAAATTAATACGACTCACTATAGG-3'), followed by 20 nt corresponding to the target-specific guide sequence and the complementary sequence (5'-GTTTTAGAGCTAGAAATAGC-3') to the 3' end of the reverse oligonucleotide (OL6137), which contain sgRNA Cas9 scaffold sequence [44]. The resulting PCR products were purified using the QIAquick PCR Purification Kit (Qiagen, Cat. 28106) and prepared in ultra-pure water at a final concentration of ~1 µg µL$^{-1}$. The DNA repair template (DRT) used to introduce gatekeeper mutations in kinases were designed as 120 nt single-stranded DNA oligonucleotide. Each template contained the desired codon mutation, silent mutations at the protospacer adjacent motif (PAM) and sgRNA binding sites to prevent re-cutting, and 20 – 30 nucleotides of homology flanking both sides of the Cas9-induced double-strand break (DSB). The single-stranded DRT were resuspended in nuclease-free water at a final concentration of 2.2 µg µL$^{-1}$. Approximately 5 µg of sgRNA and 11 µg of DRT were combined and transfected into 5x10$^6$ promastigote-stage of the *L. mexicana* cell line stably expressing T7 RNA polymerase and Cas9 endonuclease (T7/Cas9) [44]. Transfections were performed using the P3 Primary Cell 4D-Nucleofector Kit (Lonza, Cat. V4XP-3024) with a single pulse of program FI-115 in a final volume of 110 µL. Post-electroporation, cells were immediately transferred to pre-warmed HOMEM medium supplemented with 20% hi-FCS and 10 µM 6-biopterin. After a 16 hours recovery period, transfected cells were cloned into 96-well plates at a ratio of one cell per two wells.

Generation of gene knockout or endogenously tagged *Leishmania* lines was carried out using the CRISPR-Cas9 genome editing toolkit for kinetoplastids developed by Beneke et al. [44]. Primers for sgRNA and repair template construction were designed using the LeishGEdit platform (http://leishgedit.net/). For generation of null knockout mutants, two distinct repair templates – differing only in their drug resistance markers – were employed. Parasites were transfected with purified sgRNA and corresponding DNA repair templates as described above. Sixteen hours post-transfection, selective drugs (blasticidin and/or puromycin) were added, and lines were cloned by limiting dilution.

For attempted generation of catalytically inactive KKT3 mutants, we employed a drug-free CRISPR-Cas9 precision editing system using a double-stranded DNA repair template (dsDRT) [50,51]. In this approach, 144 – 200 bp dsDRTs were generated by PCR using oligonucleotides containing 20 bp complementary overhangs. The repair templates were designed to introduce the desired codon substitution, together with silent shield mutations, and comprised 38 – 63 bp homology arms flanking the Cas9-induced double-strand break. Preparation of sgRNAs and transfection of parasites were performed as previously described for single-stranded DRT-mediated genome editing.

Genomic DNA from recovered clones was extracted, and diagnostic PCRs were performed, followed by restriction enzyme digestion where required. Digestions were incubated for 16 h using the appropriate restriction enzymes and buffers recommended by New England Biolabs. Oligonucleotide sequences used to generate and validate CRISPR-Cas9-edited lines are provided in S2–S4 Tables.

Null mutants were validated by whole-genome sequencing using the Illumina platform. Paired-end reads (150 bp) were aligned to the *L. mexicana* MNYC/BZ/62/M379 Cas9/T7 reference genome [89], using Minimap2 v2.26-r1175. Read alignments were visualized with IGV v2.16.1. Genome coverage was assessed using Mosdepth v0.3.3, which calculated read depth in 500 bp windows and mean coverage across all chromosomes. Gene coverage was also determined by Mosdepth based on annotations from the MNYC/BZ/62/M379 Cas9/T7 reference. Sample coverage was normalised per chromosome, and the coverage change was calculated from the relative difference between the comparison and reference. 0.5 was used as a cut off, which equates to one copy change. The Illumina sequencing data were deposited under the SRA Bioproject accession number PRJNA1303394.

## Recombinant protein production

The gene encoding the kinase domain of *L. mexicana* CLK1 (LmxM.09.0400, residues M48 to M469) was amplified using *L. mexicana* (strain MNYC/BZ/62/M379) genomic DNA and cloned into expression vector pNIC28-Bsa4 (GenBank ID: EF198106) by ligation-independent cloning, as previously reported [40]. Analog-sensitive variants were generated by site-directed mutagenesis using the wild-type construct as a template, substituting the gatekeeper methionine residue (M213) with either alanine or glycine. All constructs were sequence-verified prior to protein expression. Expression and purification of recombinant wild-type CLK1 (rCLK1) and its analog-sensitive variants (rCLK1$^{M213A}$ and rCLK1$^{M213G}$) were performed as described previously [40]. Briefly, protein expression was carried out in *Escherichia coli* BL21(DE3)-R3-λPPase cells, followed by purification using immobilized metal ion affinity chromatography (IMAC), affinity tag removal with tobacco etch virus (TEV) protease, reverse IMAC, and a final step using size-exclusion chromatography (gel filtration – GF).

## Enzymatic assays

The enzymatic activity of rCLK1 and its analog-sensitive variants (rCLK1$^{M213A}$ and rCLK1$^{M213G}$) was assessed using commercially available kinase assays. The first assay was a homogenous time-resolved fluorescence (HTRF)-based assay (HTRF KinEASE-STK S3 kit, Cisbio #62ST3PEC), previously validated for *L. mexicana* rCLK1 [40]. Enzyme titration experiments were performed to evaluate the kinase activity of the analog-sensitive variants in comparison to the wild-type kinase. Briefly, proteins were incubated at concentrations ranging from 122 pM to 500 nM (13-point, 2-fold serial dilution) with 1 μM S3 peptide (Cisbio) and 100 μM ATP for 1 h at 30°C. All components were prepared in 1x Enzymatic Buffer (Cisbio) supplemented with 5 mM MgCl$_2$, 1 mM DTT, and 0.01% Tween-20. Reactions were performed in duplicate, and reactions with no enzyme were used as negative controls. For detection, reactions were terminated by the addition of Detection Buffer (Cisbio) containing EDTA, Streptavidin-XL665 (Cisbio) and STK antibody-Eu$^{3+}$-Cryptate (Cisbio). Fluorescence signals were measured after an additional 1 h incubation, using a CLARIOstar microplate reader (BMG LABTECH) with excitation at 330 nm, emission at 665 nm (acceptor) and 620 nm (donor). Enzyme activity was expressed as the background-corrected HTRF ratio of the acceptor and donor emission signals according to Equation [1]:

$$\text{Signal ratio (665/620)} = \left( \frac{\text{Acceptor}_{665nm}}{\text{Donor}_{620nm}} - background \right).10,000$$

(1)

where background is the background fluorescence obtained in the absence of enzyme (blank).

The second assay was a time-resolved fluorescence resonance energy transfer (TR-FRET)-based assay (LANCE Ultra Kinase Assay, Perking Elmer), used to test four different ULight-labeled peptide substrates in combination with their corresponding europium (Eu)-labeled anti-phospho-antibodies, with the aim of identifying the optimal peptide substrate for the kinases. The following peptides were evaluated: ULight-CREBtide (Ser133) (#TRF0107), ULight-Histone H3 (Thr3) (#TRF0125), ULight-4E-BP1 (#TRF0128), and ULight-GS (#TRF0131), each paired with its corresponding antibody. A

single-point selection experiment was performed using 50 nM of each peptide, 20 nM kinase, and a non-limiting concentration of ATP (100 μM). Assays were conducted according to the manufacturer's recommendations in 1x Enzymatic Buffer containing 50 mM HEPES (pH 7.5), 10 mM MgCl$_2$, 1 mM EGTA, 0.01% Tween-20 and 2 mM DTT (reaction volume = 10 μL). Reactions were performed in duplicate, and reactions with no ATP were used as negative controls. For detection, reactions were terminated after 1 h by the addition of 6 mM EDTA followed by the addition of 2 nM antibody in 1x Detection Buffer (Perkin Elmer #CR97–100) (final reaction volume = 20 μL). Fluorescence signals were measured after an additional 1 h incubation, using a CLARIOstar microplate reader (BMG LABTECH) with excitation at 320 nm and emission at 665 nm. Data acquisition was performed using MARS software (BMG LABTECH), and data analysis was carried out using GraphPad Prism v10.2.0. Enzyme activity was expressed as signal-to-background (S/B) ratio, calculated using signals obtained in the presence (+ATP) and absence (−ATP) of ATP.

### In vitro auto-phosphorylation assay

Intact mass spectrometry was used to assess the auto-phosphorylation activity of rCLK1 and its analog-sensitive variants (rCLK1$^{M213A}$ and rCLK1$^{M213G}$). Assays were performed in GF buffer (20 mM HEPES pH 7.5, 300 mM NaCl, 5% glycerol and 0.5 mM TCEP) supplemented with 5 mM MgCl$_2$ and 1 mM MnCl$_2$. Recombinant proteins (40 μM) were incubated with 1 mM ATP for 2 h at 37 °C. Proteins with no ATP were used as negative controls. Following incubation, samples were analyzed by LC-ESI-MS using a XEVO G2 Sx Q-ToF mass spectrometer (Waters) at LACTAD-UNICAMP, Brazil.

### In vitro susceptibility assay of Leishmania promastigotes

A dose response curve was set in a 96-well plate with 1x10$^4$ parasites mL$^{-1}$ treated with two-fold increasing concentrations of the bumped kinase inhibitor (1NA-PP1 or 1NM-PP1). The susceptibility of treated and untreated control was assessed after 96 hours by addition of 50 μL of 0.0125% (w/v) resazurin (Alamar Blue) prepared in PBS. Cells were incubated for an additional 2 – 4 hours at 37ºC, after which fluorescence was measured using a CLARIOstar plate reader (BMG LABTECH) with excitation at 540 nm and emission at 590 nm. Fitting of dose-response curves and IC$_{50}$ calculation were carried out using GraphPad Prism v9.3.1, with viability normalized to untreated controls (set as 100%) for each cell line.

### In vitro susceptibility assay of intracellular Leishmania amastigote

Promastigotes were cultured in HOMEM medium supplemented with 10% hi-FCS until reaching late-logarithmic phase. Differentiated BMDM were plated on 16 well Labtek tissue culture slides (Nunc, NY, USA) and then infected at a ratio of 10 promastigotes per macrophage. After 18 hours incubation at 37°C in 5% CO$_2$ using DMEM supplemented with 5% hi-FCS, non-internalized promastigotes were removed by washing, and infected macrophages were treated with serial dilutions of either BKIs 1NA-PP1 or 1NM-PP1 prepared in DMEM supplemented with 2% heat-inactivated horse serum. Following 96 hours treatment, cells were fixed with methanol and stained with Giemsa. Infected cells were quantified by light microscopy using a Zeiss Axiolab-5 microscope, with 100 macrophages counted per well to determine the percentage of infected macrophages. Parasite susceptibility was calculated as the percentage of infected macrophages in treated wells relative to untreated controls. Fitting of dose-response curves and IC$_{50}$ calculation were carried out using GraphPad Prism v10.1.0, with viability normalized to untreated controls set as 100%.

### Cytotoxicity assay

Uninfected macrophages were cultured under conditions identical to those used for the *in vitro* intracellular *Leishmania* amastigote susceptibility assay. Cells were exposed to the bumped kinase inhibitor 1NM-PP1, and cytotoxicity was quantified using a resazurin-based assay (Alamar Blue), as described above. Fluorescence readings were normalized to untreated controls (set as 100%).

## Cell viability analysis

Logarithmically growing promastigotes were cultured in the presence or absence of 1NM-PP1 (5 or 10 µM) for up to 168 h, with samples collected at 6 h and every 24 h thereafter. Following treatment, cells were stained with propidium iodide (PI) at a final concentration of 0.5 µg mL$^{-1}$ and analysed by flow cytometry using a CyAn ADP cytometer (Beckman Coulter). Data were analysed using FlowJo v10.6.2 software to quantify the proportions of PI-negative (viable) and PI-positive (non-viable) cells.

## Washout clonal recovery assay

Logarithmically growing promastigotes were cultured in the presence or absence of 1NM-PP1 (5 or 10 µM) for 6 or 24 h. Following drug exposure, cells were collected by centrifugation (1,200 x g for 10 min at 25ºC) and resuspended in fresh HOMEM medium, supplemented with 20% hi-FCS and 10 µM 6-biopterin, to remove inhibitor. Cells were then distributed into 96-well plates at a density of 48 cells per plate. After 21 days of incubation, the number of recovered clones was determined by counting wells containing viable promastigote outgrowth.

## Cell cycle analysis

Promastigote cells were cultured in the presence or absence of 5 µM 1NM-PP1 for 6 or 24 hours. Following treatment, cells were washed with PBS containing 5 mM EDTA (PBS-EDTA) and resuspended in 70% methanol. After overnight incubation at 4ºC, cells were washed once with PBS-EDTA and resuspended in 1 mL PBS-EDTA containing 10 µg mL$^{-1}$ of propidium iodide and 10 µg mL$^{-1}$ of RNase A. Following a 45-minute incubation at 37°C in the dark, DNA content was analysed by flow cytometry using a CyAn ADP cytometer (Beckman Coulter). Cell cycle distribution was determined using the Watson model in FlowJo v10.6.2 software.

## Immunofluorescence microscopy

Promastigote cells growing in the presence or absence of 5 or 10 µM 1NM-PP1 for 6 or 24 h were washed twice with PBS (1,400 g for 10 minutes at room temperature). Approximately 10$^6$ were resuspended in PBS and allowed to adhere for 15 minutes at 37°C onto poly-L-lysine-coated high-precision coverslips (thickness No. 1.5H [0.170 mm ± 0.005 mm], MARIENFELD: cat. 0107222). *In vivo* cross-linking was performed by incubating the adhered cells with 1 mM dithiobis (succinimidyl propionate) (DSP) in PBS for 10 minutes at 37°C. Cells were fixed at room temperature with 4% paraformaldehyde in PBS for 15 minutes, followed by quenching with 0.1 M glycine in PBS (pH 7.6) for 5 minutes. After two washes with PBS, cells were permeabilized with 0.5% Triton X-100 in PBS for 15 minutes. Blocking was performed by incubating cells in blocking buffer (5% BSA, 0.01% saponin in PBS) for 1 hour at room temperature. Primary immunostaining was carried out for 1 hour at room temperature using mouse anti-β-tubulin KMX-1 antibody (Sigma-Aldrich, MAB3408) diluted 1:1000 in blocking buffer. After three washes with 0.1% Triton X-100 in PBS, cells were incubated for 1 hour with Alexa Fluor 647-conjugated goat anti-mouse IgG secondary antibody (Abcam ab150119) diluted 1:1000 in blocking buffer. Following three washes with 0.1% Triton X-100/PBS, cells were counter-stained with 20 µg mL$^{-1}$ DAPI in PBS for 30 minutes, followed by a final PBS wash. Coverslips were mounted on glass slides using ProLong diamond antifade mountant (Invitrogen), according to the manufacturer's instructions.

For detection of endogenously tagged KKT2, the above protocol was followed using the following antibody combinations: mouse anti-β-tubulin KMX-1 [diluted 1:800], and rabbit anti-Myc-Tag (clone 71D10, Cell Signaling Technology mAb #2278) [diluted 1:200] as primary antibodies; and Alexa Fluor 568-conjugated goat anti-mouse IgG (Invitrogen A-11031) [diluted 1:800], and Alexa Fluor 488-conjugated donkey anti-rabbit IgG (Invitrogen A-21206) [diluted 1:200] as secondary antibodies.

## Microscopy and image analysis

Widefield fluorescence imaging was performed using a Zeiss Axio Observer7 microscope in z-stack mode (20 optical sections were captured per field). Images were processed using Fiji v2.14.0/1.54f, employing the Microvolution blind

deconvolution module. Maximum intensity projections were subsequently generated using selected z-stack layers retrieved with the "z-project" function in Fiji.

Super-resolution structured illumination microscopy (SR-SIM) was performed on a Zeiss Elyra 7 system using the Lattice SIM modality. 3D acquisition in 3 colour was performed [DAPI Excitation 405nm Emission 420–480nm; AF488 Excitation 488nm Emission 490–560nm; AF568 Excitation 561nm Emission 570–630nm] with a z-stack of 50 slices captured at 0.091 μm intervals. Zeiss Zen Black v3.0 software was used to reconstruct the images by SIM$^2$ processing using different pre-set processing parameter depending on the fluorescent signal intensities [fixed standard for β-tubulin_AF568 and KKT2_Myc_AF488; and low contrast for DAPI]. To ensure accurate colour alignment, SR-SIM images were taken of TetraSpeck fluorescent microspheres with 200 nm diameter (Thermo Fisher Scientific). Then the Zen Black Channel Alignment module was used to calculate a correction matrix that was applied to the experimental images.

### CRK9-proximal proteins affinity purification

The cell lines *L. mexicana* T7/Cas9 3xMyc::BirA*::BDF7 and 3xMyc::BirA*::CRK9 were generated using Cas9 directed endogenous tagging with pPlot BirA* Puro [44]. Cultures in 100 mL HOMEM 10% FBS were set up in quadruplicate and grown until early log stage, approximately 3x10$^6$ cells mL$^{-1}$, at this point d-biotin was supplemented to the media at 150 μM for 18 hours at 25ºC. After biotinylation cells were harvested by centrifugation (1,500 g for 10 minutes) washed twice in PBS then resuspended in pre-warmed PBS at a density of 4x10$^7$ cells mL$^{-1}$. Cells were treated with 1 mM DSP crosslinker (Thermo) for 10 minutes at 25ºC; this crosslinking was quenched with Tris-HCl pH7.5 to a concentration of 20 mM. The cells were then collected by centrifugation and stored at -80ºC until lysis. Samples were lysed with 500 μL ice cold RIPA buffer (25 mM Tris-HCl pH 7.6, 150 mM NaCl, 1% NP-40, 1% sodium deoxycholate, 0.1% SDS) containing 2x HALT protease inhibitor cocktail (Thermo) and 1x PhosSTOP (Roche). To each tube of lysate, 1μL of BaseMuncher Endonuclease (250 units, Abcam) was added and nucleic acids digested at room temperature for 10 minutes. The samples were then sonicated using a BioRuptor Pico (3 cycles at 4ºC: 30 seconds on, 30 seconds off) and clarified by centrifugation in Protein LoBind tubes (Eppendorf) at 10,000 g for 10 minutes at 4ºC. Biotinylated proteins were then enriched using 100 μL of magnetic streptavidin bead suspension (1 mg of beads, Resyn Biosciences) for each affinity purification from 4x10$^8$ parasites. Binding was performed overnight at 4ºC with end-over-end rotation. Beads were then washed in 500 μL of the following buffers for 5 min each: RIPA for 4x washes; 4 M urea in 50 mM triethyl ammonium bicarbonate (TEAB) pH 8.5; 6 M urea in 50 mM TEAB pH 8.5; 1 M KCl, 50 mM TEAB pH 8.5. Beads from each affinity purification were then resuspended in 200 μL 50 mM TEAB pH 8.5 containing 0.01% ProteaseMAX (Promega), 10 mM TCEP, 10 mM Iodoacetamide, 1 mM CaCl$_2$ and 500 ng Trypsin Lys-C (Promega). Digest was conducted overnight in a 37ºC shaking heat block, at 900 rpm. The treatment with reducing agent in this step also cleaves the DSP crosslinker. The supernatant was recovered from the beads which were then washed with 50 μL water for 5 minutes to maximise recovery of peptides. Digests were acidified with trifluoroacetic acid (TFA) to a final concentration of 0.5% and centrifuged for 10 minutes at 17,000 g to remove insoluble material. The digested peptides were desalted using Strata C$_{18}$-E columns (55 μm, 70 Å, 50 mg, 1 mL tubes – Phenomenex), elution volume was 3x 90 μL acetonitrile, peptides were dried down using a miVac Centrifugal Concentrator (Barnstead).

### Mass spectrometry data acquisition and analysis of CRK9-proximal proteins affinity purification

After Samples were loaded onto a nanoAcquity UPLC system (Waters) equipped with a PharmaFluidics μPAC C$_{18}$, Trapping column and a PharmaFluidics 50 cm μPAC C$_{18}$ nano-LC column (5 μm pillar diameter, 2.5 μM inter-pillar distance). The trap wash solvent was 0.1% (v/v) aqueous formic acid and the trapping flow rate was 10 μL min$^{-1}$. The trap was washed for 5 minutes before switching flow to the capillary column. Separation used a gradient elution of two solvents (solvent A: aqueous 0.1% (v/v) formic acid; solvent B: acetonitrile containing 0.1% (v/v) formic acid). The analytical flow rate was 1 μL min$^{-1}$ and the column temperature was 50ºC. The gradient profile was linear 2.5 – 30% B over 30 minutes

then linear 30 – 90% B over 5 minutes. All runs then proceeded to wash with 90% solvent B for 5 minutes. The column was returned to initial conditions and re-equilibrated for 5 minutes before subsequent injections.

The nanoLC system was interfaced with a maXis HD LC-MS/MS system (Bruker Daltonics) with CaptiveSpray ionisation source (Bruker Daltonics). Positive ESI-MS and MS/MS spectra were acquired using MRM mode to define data independent acquisition (DIA) windows with a width of 5 Th between $m/z$ 450 – 650. Instrument control, data acquisition and processing were performed using Compass 1.7 software (microTOF control, Hystar and DataAnalysis, Bruker Daltonics). Instrument settings were: ion spray voltage: 1,450 V, dry gas: 3 L min$^{-1}$, dry gas temperature 150ºC, ion acquisition range: $m/z$ 280 – 1,600, spectra rate: 15 Hz, quadrupole low mass: 322 $m/z$, collision RF: 1,400 Vpp, transfer time 120 ms. The collision energy was set to 22 for DIA windows below m/z 600 and to 24 for higher m/z windows.

LC-MS data, in Bruker.d format, were converted to.mzML format using MSConvert (ProteoWizard) before analysing using DIA-NN (1.8.1) with searching against and in-silico predicted spectral library, derived from the LmexCas9T7-prot database appended with common proteomic contaminants. Search criteria were set to maintain a false discovery rate (FDR) of 1%. Peptide-centric output in.tsv format, was pivoted to protein-centric summaries using KNIME and data filtered to require protein q-values < 0.01 and a minimum of two peptides per accepted protein. Protein intensities were log2 transformed and proximal proteins were determined with the limma package [56] using options trend = TRUE and robust = TRUE for the eBayes function. Protein intensities in CRK9 samples were compared to those in the spatial reference BDF7 to determine proximal proteins. Multiple testing correction was carried out according to the Benjamini-Hochberg procedure, the false discovery rate for identified proximals was 1%.

## SM1–71 affinity purification

Cell pellets (in triplicate for each sample) of 3x10$^8$ parasites washed with PBS were used for each affinity purification and lysed in 300 µL ice-cold lysis buffer (0.1% sodium dodecyl sulphate (SDS), 0.5% sodium deoxycholate, 1% IGE-PAL CA-630, 0.1 mM EDTA, 125 mM NaCl, 50 mM Tris-HCl pH 7.5) containing 0.1 mM 4-(2-aminoethyl)benzenesulfonyl fluoride, 1 µg mL$^{-1}$ pepstatin A, 1 µM E-64 and 0.4 mM 1–10 phenanthroline. In addition, every 10 mL of lysis buffer was supplemented with 100 µL protease inhibitor cocktail (abcam) plus 1 tablet of cOmplete protease inhibitor cocktail and PhosSTOP (Roche). Cells were lysed using a Bioruptor sonicator (Diagenode) for 3 cycles (30s on/off) in 1.5 mL microtubes containing silica beads to aid shearing. 1 µL of BaseMuncher endonuclease (250 Units, abcam) was added to each lysate and incubated on ice for 1 hour to digest nucleic acids. Lysates were clarified by centrifugation at 10,000 g for 10 minutes at 4ºC and the supernatant transferred to LoBind Eppendorf microtubes. Lysates were pre-treated with SM1–71 (final concentration 4 µM) or DMSO (to 0.2%) for 1 hour before adding SM1–71-biotin (4 µM) for 2 hours. For enrichment of biotinylated material, 100 µL of magnetic streptavidin bead suspension (1 mg, protein binding capacity ≥ 0.3 mg/mL IgG - Resyn Biosciences) was used to affinity purify by end-over-end rotation at 4ºC overnight. Beads with bound proteins were washed in 500 µL of the following for 5 minutes each: lysis buffer (plus protease inhibitors); lysis buffer; PBS supplemented with 0.025% Tween-20 then PBS/Tween supplemented with 2% SDS (55ºC) before rinsing with PBS/Tween and transferring the beads to clean LoBind tubes. Further washes using 4 M urea, 6 M urea, 1 M KCl and 50 mM TEAB pH 8.5 were performed and the beads stored at -20ºC.

## Mass spectrometry data acquisition and analysis of SM1–71 affinity purification

After rinsing with 50 mM TEAB, proteins were denatured, reduced and enzymatically digested on-bead with the addition of 100 µL of 50 mM TEAB containing the following: 0.01% (w/w) ProteaseMAX surfactant (Promega); 2.9 mg mL$^{-1}$ tris(2-carboxyethyl)phosphine; 100 mM CaCl$_2$ and 2.5 µg sequencing grade trypsin/Lys-C mix (Promega). Beads were shaken for 5 minutes at 850 rpm before incubation overnight at 37ºC. Supernatant containing peptide was acidified to contain 0.5% trifluoracetic acid before spinning at 17,000 g for 10 minutes to pellet precipitated ProteaseMAX. A 40 µL aliquot of

soluble material was taken per sample for LC-MS acquisition. Resulting peptides were desalted with $C_{18}$ ZipTip (0.2 uL, Millipore) before being re-suspended in aqueous 0.1% (v/v) trifluoroacetic acid.

Peptides were loaded onto an mClass nanoflow UPLC system (Waters) equipped with a nanoEaze M/Z Symmetry 100 Å, $C_{18}$, 5 µm trap column (180 µm x 20 mm, Waters) and a PepMap, 2 µm, 100 Å, $C_{18}$ EasyNano nanocapillary column (75 µm x 500 mm, Thermo). The trap wash solvent was aqueous 0.05% (v/v) trifluoroacetic acid and the trapping flow rate was 15 µL min$^{-1}$. The trap was washed for 5 minutes before switching flow to the capillary column. Separation used gradient elution of two solvents: solvent A, aqueous 0.1% (v/v) formic acid; solvent B, acetonitrile containing 0.1% (v/v) formic acid. The flow rate for the capillary column was 300 nL min$^{-1}$ and the column temperature was 40ºC. The linear multi-step gradient profile was: 3 – 10% B over 7 minutes, 10 – 35% B over 30 minutes, 35 – 99% B over 5 minutes and then proceeded to wash with 99% solvent B for 4 minutes. The column was returned to initial conditions and re-equilibrated for 15 minutes before subsequent injections.

The nanoLC system was interfaced with an Orbitrap Fusion Tribrid mass spectrometer (Thermo) with an EasyNano ionisation source (Thermo). Positive ESI-MS and MS$^2$ spectra were acquired using Xcalibur software (version 4.0, Thermo). Instrument source settings were: ion spray voltage, 1,900 V; sweep gas, 0 Arb; ion transfer tube temperature; 275ºC. MS$^1$ spectra were acquired in the Orbitrap with: 120,000 resolution, scan range: $m/z$ 375 – 1,500; AGC target, 4e$^5$; max fill time, 100 ms. Data dependant acquisition was performed in top speed mode using a 1 s cycle, selecting the most intense precursors with charge states >1. Easy-IC was used for internal calibration. Dynamic exclusion was performed for 50 s post precursor selection and a minimum threshold for fragmentation was set at 5e$^3$. MS$^2$ spectra were acquired in the linear ion trap with: scan rate, turbo; quadrupole isolation, 1.6 $m/z$; activation type, HCD; activation energy: 32%; AGC target, 5e$^3$; first mass, 110 $m/z$; max fill time, 100 ms. Acquisitions were arranged by Xcalibur to inject ions for all available parallelizable time.

LC-MS chromatograms in.raw format were imported into Progenesis QI (Waters, v.3.3) for peak picking and alignment. A concatenated MS$^2$ peak list was exported and searched against the *L. mexicana* subset of the TryTripDB database (v.54) appended with common proteomic contaminants using Mascot Server (Matrix Science v.2.5). Database search criteria specified: Enzyme, trypsin; Max missed cleavages, 1; Variable modifications, Oxidation (M); Peptide tolerance, 3 ppm; MS/MS tolerance, 0.5 Da. Peptide identifications were filtered through the Percolator algorithm to achieve a false discovery rate of 1% as assessed empirically against a reversed database search. Peptide identifications were reimported into Progenesis QI and associated with precursor intensity signals and identifications mapped between runs. Relative peptide abundance was obtained by integrating areas under identified MS$^1$ signals and inference from peptide values. Normalisation was applied between runs on the basis of total observed peptide signal. Protein quantifications were filtered to require a minimum of two peptides per protein. A multi-way ANOVA was applied within Progenesis QI to call differing abundance with the Hochberg and Benjamini multiple test correction applied and q < 0.05 set as significance level.

**Identification of CLK1 modification by SM1–71 using LC-MS/MS**

To identify CLK1 residues forming covalent adducts with SM1–71, the compound was incubated with purified rCLK1 at an inhibitor-to-protein ratio of 3:1 for 2 h at room temperature. Untreated protein was used as a control. Quadruplicate samples were partially resolved by SDS-PAGE. Protein was in-gel digested with trypsin (Promega) following reduction with DTE and alkylation with iodoacetamide. Peptides were loaded onto EvoTip Pure tips for desalting and as a disposable trap column for nanoUPLC using an EvoSep One system. A pre-set EvoSep 100 SPD gradient (from Evosep One HyStar Driver 2.3.57.0) was used with an 8 cm EvoSep C18 Performance column (8 cm x 150 mm x 1.5 mm). The nanoUPLC system was interfaced to a timsTOF HT mass spectrometer (Bruker) with a CaptiveSpray ionisation source (Source). Positive PASEF-DDA, ESI-MS and MS$^2$ spectra were acquired using Compass HyStar software (version 6.2, Bruker). Instrument source settings were: capillary voltage, 1,400 V; dry gas, 3 L/min; dry temperature; 180°C. Spectra were acquired between m/z 100–1,700. TIMS settings were: 1/K0 0.6-1.60 V.s/cm$^2$; Ramp time, 100 ms; Ramp rate 9.42 Hz. Data dependent

acquisition was performed with 10 PASEF ramps and a total cycle time of 1.17 s. An intensity threshold of 2,500 and a target intensity of 20,000 were set with active exclusion applied for 0.4 min post precursor selection. Collision energy was interpolated between 20 eV at 0.6 V. s/cm$^2$ to 59 eV at 1.6 V. s/cm$^2$.

Acquired spectra were searched using MSFragger version 4.4 running under Fragpipe version 23.1. Searching was run with a 1% FDR target achieved with the percolator algorithm. Variable modification was set for Cys residues as +57.0215 Da for iodoacetamide (IAM) modification and +463.1888 for SM1–71 modification.

## Statistics

Data were collected from at least three independent experiments, unless otherwise indicated. Statistical analyses were performed using GraphPad Prism v10.2.0. The appropriate tests were conducted and are detailed in the corresponding figure legends. Results are presented as the mean ± standard error of mean (SEM).

## Supporting information

**S1 Table. Protein kinase genes previously identified by our group as required in *L. mexicana* promastigotes.**
(PDF)

**S2 Table. Sequence of oligonucleotides used to generate sgRNA and DNA repair template for CRISPR-Cas9 edited lines.**
(PDF)

**S3 Table. Sequence of the DNA repair templates used to engineer analog-sensitive kinases in *Leishmania*.**
(PDF)

**S4 Table. Sequence of oligonucleotides used to screen the CRISPR-Cas9 engineered *L. mexicana* cell lines.**
(PDF)

**S1 Fig. Schematic illustration to identify protein kinases required for *L. mexicana* amastigote survival.**
(PDF)

**S2 Fig. Gatekeeper residues in the *L. mexicana* kinome.**
(PDF)

**S3 Fig. Conservation of CLK1, KKT2, KKT3 and CRK9 across trypanosomatids.**
(PDF)

**S4 Fig. CRISPR-Cas9-mediated engineering of analog-sensitive KKT2 in *Leishmania*.**
(PDF)

**S5 Fig. CRISPR-Cas9-mediated engineering of analog-sensitive KKT3 in *Leishmania*.**
(PDF)

**S6 Fig. CRISPR-Cas9-mediated engineering of analog-sensitive CRK9 in *Leishmania*.**
(PDF)

**S7 Fig. CRISPR-Cas9-mediated engineering of analog-sensitive CLK1/CLK2 in *Leishmania*.**
(PDF)

**S8 Fig. Purification and substrate recognition of recombinant CLK1 and its analog-sensitive variants.**
(PDF)

**S9 Fig. Susceptibility of *L. mexicana* lines expressing analog-sensitive variants of KKT2 and KKT3 to 1NA-PP1.**
(PDF)

**S10 Fig. Growth curves of *L. mexicana* analog-sensitive kinase mutants and parental line.**
(PDF)

**S11 Fig. Requirement of KKT3 protein kinase for *L. mexicana* promastigote survival.**
(PDF)

**S12 Fig. Engineering KKT3 kinase-dead mutants via targeted mutation of the three catalytic residues.**
(PDF)

**S13 Fig. Engineering KKT3 kinase-dead mutants by deletion of the kinase domain.**
(PDF)

**S14 Fig. Growth kinetics of *L. mexicana* KKT3 kinase dead mutants and parental line.**
(PDF)

**S15 Fig. Domain architecture of *L. mexicana* KKT3.**
(PDF)

**S16 Fig. Effect of KKT2 kinase activity inhibition on *Leishmania* cell cycle progression.**
(PDF)

**S17 Fig. Engineering of the KKT2_AS.mNG.3xMyc cell line.**
(PDF)

**S18 Fig. Spatial and temporal distribution of *L. mexicana* KKT2 throughout the cell cycle.**
(PDF)

**S19 Fig. Effect of CRK9 kinase activity inhibition on *Leishmania* cell cycle progression.**
(PDF)

**S20 Fig. WZ8040 and SM1–71 probes.**
(PDF)

**S21 Fig. Predicted structural models of kinase domains from *L. mexicana* protein kinases enriched by the multi-targeted acrylamide-modified probe, SM1–71-biotin.**
(PDF)

**S22 Fig. Phylogenetic analysis of human CDKs with sequence similarity to *L. mexicana* CRK9.**
(PDF)

**S1 Data. Mapping of gatekeeper residues across the *L. mexicana* kinome.**
(XLSX)

**S2 Data. Identification of CRK9-proximal proteome.**
(XLSX)

**S3 Data. Identification of protein kinases with targetable cysteines by the multi-targeted SM1–71-biotin probe.**
(XLSX)

## Acknowledgments

We thank our colleagues in the Bioscience Technology Facility of University of York who provided insight and expertise that greatly assisted our microscopy, flow cytometry, genomic data analysis, and mass spectrometry research. The York Centre of Excellence in Mass Spectrometry was created thanks to a major capital investment through Science City York, supported by Yorkshire Forward with funds from the Northern Way Initiative, and subsequent support from EPSRC (EP/K039660/1; EP/M028127/1).

## Author contributions

**Conceptualization:** Juliana B. T. Carnielli, James A. Brannigan, Rafael M. Couñago, Peter Sjö, Ana Paula C. A. Lima, Anthony J. Wilkinson, Jeremy C. Mottram.

**Data curation:** Juliana B. T. Carnielli.

**Formal analysis:** Juliana B. T. Carnielli, James A. Brannigan, Priscila Z. Ramos, Nathaniel G. Jones, Ana Paula C. A. Lima.

**Funding acquisition:** Ana Paula C. A. Lima, Anthony J. Wilkinson, Jeremy C. Mottram.

**Investigation:** Juliana B. T. Carnielli, James A. Brannigan, Priscila Z. Ramos, Nathaniel G. Jones, Rafael M. Couñago, Peter Sjö, Anthony J. Wilkinson.

**Methodology:** Juliana B. T. Carnielli.

**Supervision:** Anthony J. Wilkinson, Jeremy C. Mottram.

**Visualization:** Juliana B. T. Carnielli, James A. Brannigan, Priscila Z. Ramos, Nathaniel G. Jones.

**Writing – original draft:** Juliana B. T. Carnielli.

**Writing – review & editing:** Juliana B. T. Carnielli, James A. Brannigan, Priscila Z. Ramos, Nathaniel G. Jones, Rafael M. Couñago, Peter Sjö, Ana Paula C. A. Lima, Anthony J. Wilkinson, Jeremy C. Mottram.

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
