## [Decision Letter · Decision Letter 0]

3 Nov 2025

PPATHOGENS-D-25-02309

Chemical genetics reveals Leishmania KKT2 and CRK9 kinase activity is required for cell cycle progression

PLOS Pathogens

Dear Dr. Brambilla Trindade Carnielli,

Thank you for submitting your manuscript to PLOS Pathogens. After careful consideration, we feel that it has merit but does not fully meet PLOS Pathogens's publication criteria as it currently stands. Therefore, we invite you to submit a revised version of the manuscript that addresses the points raised during the review process.

Please submit your revised manuscript within 60 days Jan 02 2026 11:59PM. If you will need more time than this to complete your revisions, please reply to this message or contact the journal office at plospathogens@plos.org. Please include the following items when submitting your revised manuscript:

We look forward to receiving your revised manuscript.

Kind regards,

Michael Boshart

Academic Editor

PLOS Pathogens

Tracey Lamb

Section Editor

PLOS Pathogens

Sumita Bhaduri-McIntosh

Editor-in-Chief

PLOS Pathogens

orcid.org/0000-0003-2946-9497

Michael Malim

Editor-in-Chief

PLOS Pathogens

orcid.org/0000-0002-7699-2064

**Additional Editor Comments:**

This is technically advanved and very interesting work potentially providing a valuable ressource for researchers interested in parasite kinases and drug target identification. Three expert reviewers agree that the value of this ressource should be improved by essential validation and control experiments as detailed in the reviews, addition of methods details and references and clarifications in the text.

All reviewers agree that the analog-sensitive kinase screen and the search for covalently adressable kinase domains are rather disconnected parts. As the second part is not validated and would definitely require significant additional experiments to be acceptable, its separate publication with additional data would be a good option.

Of major concern in the first part are the absence of mechanistic insight into the inhibition (enzyme kinetics of mutant kinases, IC₅₀ for BKIs) and the functional consequences of kinase inhibition. Authors should provide biochemical characterization of mutant kinases and experiments to connect the screening results with the interaction proteomics datasets as suggested by reviewer 3.

**Journal Requirements:**

1) We do not publish any copyright or trademark symbols that usually accompany proprietary names, eg ©,  ®, or TM  (e.g. next to drug or reagent names). Therefore please remove all instances of trademark/copyright symbols throughout the text, including:

- ® on page: 34.

- TM on page: 34 and 35.

**Reviewers' Comments:**

Reviewer's Responses to Questions

**Part I - Summary**

Reviewer #1: As stated in their title “Chemical genetics reveals Leishmania KKT2 and CRK9 kinase activity is required for cell cycle progression” Carnielli et al investigate these two kinases using a chemical genetics approach. They generated analogue-sensitive kinases using CRISPR-Cas9 precision genome editing technology for KKT2, CRK9 and KKT3 employing a novel marker-free methodology. Only parasites expressing KKT2 and CRK9 AS versions showed proliferation defects when treated with specific inhibitors in promastigote cultures and in infected macrophages. The authors narrowed down the role of each of these kinases in the cell cycle with mitotic arrest at a stage prior to spindle assembly for KKT2, and impaired DNA synthesis for CRK9, resulting in cell death. For detailed analysis a KKT2_AS::mNG::3xMyc cell line using blasticidin as a selectable marker was generated to attempt localization of the kinase during the cell cycle. The interactome for CRK9 using 3xMyc::BirA*::CRK9 was determined to 137 proximal proteins. Then the authors switch gears and use SM1-71-biotin, a covalent broad-spectrum kinase inhibitor linked to biotin to enrich for kinases with cysteine residues in an accessible position identifying 29 from Leishmania mexicana, which they could further narrow down to 10 kinases using a competitive chemoproteomic approach. Taken together the manuscript is a very well written compilation of highly interesting methodologies but lacks a hypothesis driven approach and research question.

Reviewer #2: This is a well-written paper describing the implementation of a chemical genetics approach to investigate the function of selected protein kinases (PKs) of the protistan parasite Leishmania. The study builds on a recent Nature Comms paper by the same team where essential kinases were identified by kinome-wide reverse genetics. The barcode-based scoring system used here to identify PKs that are required for amastigote survival is elegant. The AS approach undertaken here to fully validate the selected kinases as targets is elegant as well, the data are compelling and statistically supported, and their interpretation perfectly sound. It is noteworthy that this approach allows to interrogate PK functions in amastigotes, which has been a significant hurdle.

Overall, this study represents a clear advance in parasite functional kinomics, in our understanding of cellular regulation through phosphosignalling in Leishmania (notably with respect to cell cycle control), and it the validation of potential targets for chemotherapeutic intervention. Indeed, in view of the amply-documented draggability of protein kinases, validation of targets clearly paves the way to PK-direct antiparasitic drug discovery/development programmes. A strategy focussing on Cys-dependent covalent inhibition of selected PKs is particularly attractive in this respect.

I have no major issue with the manuscript, and I have no doubt this will be of great interest to the readership of PLoS Pathogens. However, the authors might want to consider the points listed below.

Reviewer #3: The manuscript “Chemical genetics reveals Leishmania KKT2 and CRK9 kinase activity is required for cell cycle progression” by Carnielli et al., explores the drug-target potential of Leishmania protein kinases through two approaches: (i) the generation of analog-sensitive (AS) kinases by mutating the gatekeeper residue, thereby allowing selective inhibition with bulky ATP-competitive analogues (1NM-PP1 or 1NA-PP1); and (ii) the identification of kinases that contain targetable cysteines, leading to covalent inhibition. Although both strategies are presented, the manuscript does not explain why they are pursued together or how they reinforce each other, leaving the overall logic of the study unclear.

Building on their previous work (Baker et al.) that used CRISPR Cas9 to identify essential protein kinases throughout the Leishmania life cycle, the first part of this study proposes an additional tool to characterise those essential enzymes. Four kinetoplastid-specific kinetochore kinases, CLK1 (KKT10), CLK2 (KKT9), KKT2, KKT3, were selected as examples because they are divergent from human kinases and thus constitute attractive drug targets. Gatekeeper mutations were introduced into these kinases, generating AS versions that can be selectively inhibited. The authors presented mainly a phenotypic characterization of two mutants, KKT2-AS and CRK9-AS, showing that the chemical inhibition of each kinase impairs cell cycle progression. For CRK9, the authors also mapped its proximal proteome, the GO term enrichment confirmed its role in transcriptional regulation.

This first part is noteworthy because it tackles the challenging problem of probing essential protein kinases in Leishmania, a parasite in which conventional genetic tools such as inducible promoters or RNA‑i are largely unavailable. By focusing on KKT2, a kinetochore component involved in chromosome segregation, the authors extend our understanding of the cell‑cycle machinery in this organism. Nonetheless, the study provides only phenotypic observations without proposing mechanistic insights into the regulation of cell cycle progression by KKT2 (see major). Although the authors previously mapped the KKT2 proximal proteome and identified several putative interactors (Ref 7), these data are not exploited in the current manuscript to add to the initial observations. For example, the authors could have examined whether KKT2’s kinase activity is required for the proper localization of selected proximal proteins. Similarly, the data on CRK9 are largely confirmatory. Although the authors performed a proximity‑labelling experiment to uncover potential CRK9 interactors, these results are presented only as a list without any functional validation or link with the CRK9-AS experiment. For example, they could have checked whether inhibition of CRK9-AS change the proximity proteome, with the loss of some proteins. While the work successfully confirms the importance of KKT2 and CRK9 in cell cycle, the proteomic datasets were not exploited to highlight the molecular mechanisms by which these kinases control cell‑cycle progression and gene expression.

The second part of the study focuses on identifying protein kinases that contain targetable cysteine residues, with the goal of enabling the design of covalent inhibitors targeting these kinases. Using SM1-71, a compound that occupies the ATP-binding pocket and forms a covalent bond with a proximal cysteine, the authors screened the kinome, and initially identified 29 kinases, 14 of which have a cysteine positioned near the DFG motif. A subsequent competitive chemoproteomic assay was employed to enrich for kinases that engage SM1-71 with higher specificity, narrowing the list to 11 candidates (10 of which overlapped with the first screen). While the authors provide a comprehensive catalog of these cysteine-containing kinases, the manuscript does not present any experimental validation of target engagement (e.g., thermal-shift assays, pulldown experiments). Consequently, this section remains largely descriptive and does answer whether and how these kinases might be exploited as therapeutic targets.

In summary, the work delivers a valuable “genetic chemical” toolbox for interrogating essential kinases in Leishmania and identifying drug targets ready for covalent inhibition. To strengthen the manuscript, the authors should more clearly articulate their manuscript, provide some mechanistic data for the highlighted kinases.

**Part II – Major Issues: Key Experiments Required for Acceptance**

Reviewer #1: The section, Leishmania kinome, a source of attractive drug targets is not really required for the investigation described in the title. Nevertheless, I include observations related to its content as they are important to rectify mistakes made earlier.

Lines 169-172 Cytoplasmic localization in promastigotes does not necessarily mean the protein is also expressed in amastigotes and found in the same compartment.

There appears to be something wrong with MKK1 and MKK4 in the Baker et al paper Nat Commun 12, 1244 (2021), which seems to carry over into this manuscript. LmxM.08_29.2320 MKK1 is shown with a red dot in the kinase tree of figure 1 a and hence is labelled as “could not be deleted” by Baker et al using CRISPR-Cas9 technology. In figure 3 c in Baker et al a deletion of MKK1 in the presence of a trans gene is documented. Was that the cell line that was used in the pool with the null mutants of the other kinases in this manuscript? It remains unclear whether Baker et al could not generate a null mutant for MKK1 or whether a null mutant for MKK1 did not show a motility defect. MKK1 could be deleted in a previous publication, EUKARYOTIC CELL, 2003, 769–777 and the null mutant was found to be able to cause lesions in Balb/c mice and differentiate back into promastigotes. Moreover, MKK1 (LmxMKK) was shown to be absent in lesion-derived amastigotes of L. mexicana wild type using immunoblot analysis with an MKK1 specific antiserum.

For noting: In table 1, the Baker et al paper claims that LmxM.24.2320 MKK4 (LmxPK4) could not be deleted using CRISPR-Cas9 technology. However, this kinase was not labelled with a red dot in figure 1. Again, in a previous publication Molecular Microbiology (2005) 56(5), 1169–1182, null mutants for this kinase could be generated using homologous recombination. In case the authors decide to correct their previous publication, they should also check their references for correct numbering, occurrence and fit.

Table 1 of the manuscript under review shows PK4 as LmxM.21.1650. Please check whether this should be LmxM.24.2320 or a different name.

Information on null mutants of MPK5 and MPK12 has been published in the International Journal for Parasitology 37 (2007) 1053–1062. MPK5 null mutants persist in infected mice without lesion development and MPK12 null mutants are as infective as L. mexicana wild type. Additionally, LmxMPK7 genomic null mutants carrying an episomal copy of the kinase gene lost this copy when cultured as promastigotes in the absence of selective pressure or when used to infect Balb/c mice (Bleicher N, PhD thesis, Hamburg, https://ediss.sub.uni-hamburg.de/handle/ediss/8014). The virulence of LmxMPK7 null mutants was significantly reduced causing delayed lesion development or no lesions at all. However, using null mutants obtained from mouse lesions led to normal lesion development in a subsequent infection without delay suggesting compensation. This suggests that MPK7 might not be the best drug target.

I assume the final time point in the barcode experiment is 6 weeks post infection. Please, explain why a 50% decrease is sufficient to assume that the respective kinase is essential for survival of the amastigotes. Would you not expect to see a complete loss of those mutants having lost a kinase that is essential?

“A protein kinase was classified as required for amastigote survival if the corresponding mutant cell line exhibited the following trajectory in barcode representation: (i) a significant ≥2-fold reduction in barcode abundance at the first post infection time point, followed by an additional decrease of ≥30% at the second time point (meaning ≥0.5 at first time point followed by ≥0.35 at second); or (ii) no significant change in the first time point, but a significant ≥50% reduction at the final time point (meaning ≥0.5 at last time point, but still the previous time point being higher) (Supplementary Fig 1).” Might not be sufficiently stringent.

Also, replace “≥2-fold reduction” with “≥50% reduction” for consistency.

It is not clear how the above criteria were used to analyse the data presented in supplementary data 1. CLK1, CLK2, KKT2, and KKT3 are shown as not present in the library, hence, no ratios can be determined but they are still labelled as “Required” in macrophages. In addition, why are LmxM.08_29.2020, LmxM.36.0720, and LmxM.34.4620 not required in macrophages? They seem to fulfil the criteria. Likewise, to name a few others, for LmxM.19.1470, LmxM.22.0810, and LmxM.04.1210 it is not clear why they are “Required” based on the data for the situation in macrophages. It looks like that the trajectory from FP3 to FP6 for LmxM.04.1210 is pointing upwards rather than down. I might misinterpret the numbers. To avoid this from happening a better explanation is needed.

To conclude, this section needs to be corrected and much improved or removed in its entirety, as the analysis does not lead to the selection of the four mentioned kinases, nor CRK9.

The section on checking whether Leishmania might be suitable for testing using bumped kinase inhibitors does not warrant that much space and could possibly reduced to a few sentences.

Line 286, check the Jones NG publication. There is no CRK9 mentioned in the cited paper. CRK9 AS data for promastigotes should be included.

In line 323 onwards, the authors argue that KKT3 is required in Leishmania but not for its kinase activity on the basis that the AS mutant version does not show an effect in cells treated with the inhibitor. There is no evidence provided that KKT3 AS is actually sensitive to the inhibitor. This approach does not work for all kinases. Unfortunately, KKT3 is a big molecule but recombinant expression of the kinase domain (amino acids 1-337) might be possible for the wild type and AS mutants followed by enzymatic assays to assess whether the AS mutant can be inhibited. The authors mention further experiments are required later in line 545 onwards.

Figure 4 d requires a wild type control for the myc tag or better even an immunoblot showing that the anti-myc antibody used here only recognises the tagged KKT2.

Reviewer #2: No major issue.

Reviewer #3: - The authors should evaluate recombinant versions of the “bumped” kinases in an in-vitro kinase assay to verify their direct inhibition by 1NM-PP1 and 1NA-PP1. These experiments would clarify the mechanistic differences between the two analogues and rule out the possibility that the absence of a phenotype for KKT3 is simply due to ineffective inhibition of its catalytic activity. Demonstrating that KKT3 kinase activity can be blocked without affecting cell viability would provide strong support for the authors claim that “KKT3, but not its kinase activity, is essential for promastigote viability.”

- For the untreated control, the infection rate (percentage of infected macrophages) and the parasite burden (number of parasites per 100 macrophages) measured at different time points up to 96 hours (every 24h) for the parental strain and the strains containing the bumped kinases should be presented. This time-course data will allow a clear assessment of parasite survival, differentiation and multiplication in the absence of any treatment and determine whether the kinetic of infection is comparable between the parental strain and the strains containing the bumped kinases.

- The authors should investigate whether the tested compounds have any activity on mouse macrophage proteins, particularly host kinases that might be naturally sensitive, to determine if off-target effects on the host cells could contribute to the observed differences in IC₅₀ values between promastigotes and amastigotes.

- The authors identified several kinases that contains targetable cysteines (Table 3). They should validate their results by selecting at least one kinase and perform a target engagement assay, such as cellular thermal shift assay (CETSA), with SM1-71 to demonstrate covalent inhibition.

**Part III – Minor Issues: Editorial and Data Presentation Modifications**

Reviewer #1: Line 227 should be "related trypanosomatid". Delete “As” at the beginning of the sentence in line 400.

Reviewer #2: The authors might want to consider the points listed below.

Line 61: parasites (plural)

Line 62: “the mechanisms of action of most antileishmanial drugs remain incompletely understood.” I would suggest mentioning the mechanisms of resistance here, as the sentence later mentions drug resistance surveillance strategies; mechanisms of action and of resistance are often uncoupled (e.g efflux transporters). Suggestion: the mechanisms of action of most antileishmanial drugs, and the mechanism underlying resistance to these compounds, remain incompletely understood.

Line 118 ff: The idea of implementing a gatekeeper-based chemical genetics analysis for protein kinases of parasites has been proposed two decades ago (PMID: 12377287; one of the authors is the senior author of the present manuscript. It would be appropriate to cite this paper at the start of the paragraph in line 118.

Line 136 ff: In a similar vein, the targeting of Cysteine residues in PKs has also been explored with success in Plasmodium, another major protistan parasites, and, interestingly, in a PK of the CLK family (e.g. PMID: 39441986). It would be relevant to briefly mention this in the introduction, may at line 137, after citing Ref [6].

Line 199 ff: “Of the 193 predicted ePKs previously annotated in the L. mexicana kinome [14], 10 are classified as pseudokinases as they lack all three of the conserved catalytic residues (K D D) required for kinase activity (Fig 1a). Given their lack of enzymatic function and demonstrated non-essentiality, pseudokinases were not analysed for gatekeeper residue identity.” There has been debate about the total inactivity of pseudokinases (e.g. PMID: 20421461). How onerous would it be to check the gatekeeper of these 10 sequences labelled as pseudokinases? It would not hurt to include them.

The BioID interactomics study performed for CRK9 yielded interesting results; was this also considered for KKT2?

Line 506ff: “This finding stands in contrast to several apicomplexan parasites, including Toxoplasma gondii [50], Cryptosporidium parvum [51], and Neospora caninum [52], where glycine residues are found at the gatekeeper position in calcium dependent protein kinase 1 (CDPK1), thereby enabling selective inhibition by bumped kinase inhibitors and supporting their development in a targeted therapeutic approach.” It may be interesting to briefly mention that phenotypic/functional studies could indeed be implemented for some of the kinases with natural small gatekeepers, by generating mutant with a large gatekeeper, and comparatively evaluating the biochemical response to BKIs (to which the mutant are insensitive). See for example the papers on Plasmodium PKG and the T618Q gatekeeper substitution (e.g PMID PMID: 19915077; 38372781).

Line 526. “We were unable to generate CLK1/CLK2 AS mutants, probably because the mutation is not tolerated by the kinase”. It has been shown that the gatekeeper Gly or Ala mutation can severely impair kinase activity. In many cases, this can be rescued by introducing a compensatory mutation, that restores activity and maintains susceptibility to BKI (see PMID: 30801966 for a review). This may deserve a comment in the discussion as a possible way forward to assess the function of CLK1/2 (see my comment above about the use of recombinant kinases to test the effect of the Gatekeeper mutations --this could ascertain rescue by the secondary mutation, CLK1/2 are shown to be rendered inactive by the gatekeeper mutation).

Lines 539 ff: It is interesting that the KKT3 AS mutation did not have any effect on the viability of parasite upon BKI treatment, but that the gene cannot be deleted (or knocked-down) entirely. It would be of interest to briefly discuss if there are other recognisable domains in addition to the kinase domain (which can be deleted) on the KKT3 polypeptide, whose deletion would be lethal.

Line 543: “Here, we cannot exclude the possibility that the gatekeeper residue substitution in AS-KKT3 failed to sufficiently expand the ATP-binding pocket to permit effective binding of bumped kinase inhibitors”. It might indeed be the case that BKI cannot access the catalytic pocket even in the gatekeeper mutant. Secondary mutations have been in some case been shown to sensitise the mutated kinase to BKI inhibition --this could also deserve a comment (see the same paper as mentioned in the comment above, PMID: 30801966).

Overall comment on the two aspects of the manuscript. The two aspects of the study (1: Validation of PKs by chemical genetics and 2: search for cysteine-containing kinase domains) seem a bit disconnected, and do not support each other in terms of data. The authors might want to write a separate paper focussing on the cysteine-targeting story. Alternatively, an additional paragraph in the discussion outlining a similar AS chemical genetic approach for MPK4, MPK5, MPK7, and AEK1, that harbour reactive cysteine residues amenable to covalent inhibition, may improve the general self-coherence of the manuscript. One (maybe wild) suggestion: could it be that CLK1/2 were refractory to the AS mutation be linked to the presence of a cysteine in the catalytic domain? It may be worth it investigating the literature (e.g. the paper cited above, PMID: 30801966) to determine if cysteine-containing PKs are over-represented in the subset of PKs that required a secondary mutation to implement the AS gatekeeper chemical genetics approach.

Overall comment on the use of recombinant kinases: in line with previous comment, and in relation to the discussion about fitness of the parasites carrying an AS mutation (line 310), it would be nice to measure, using recombinant kinases, (i) the enzymatic activity parameters of the mutant kinases (in relation to the WTs), and (ii) the shift in IC50 for the BKIs. This is not a requirement, as the controls used throughout the study (effect of the BKIs on wild-type parasites and mock-treatment of the AS mutant) are sufficient to assign the phenotypic effects to the PK of interest, KKT3 and CRK9. However, with respect to CLK1/CLK2 (for which AS mutants could not be obtained), it might provide an answer to the question of whether this lack of success is caused by an impairment of kinase activity because of the mutation (see also above my comment on Line 526)

Reviewer #3: - The authors should provide the list of the 43 essential protein kinases they previously identified, as a supplementary table, thereby making the information more readily accessible. They should also mention that KKT2 and KKT3 were selected from the 43 essential protein list, while CLK1/2 were selected from Table 1, protein kinases only essential in amastigotes.

- The authors should provide, as supplementary material, alphafold models of CLK1/2 and KKT3 both with and without the glycine/alanine residues. This will allow evaluation of whether the resulting conformational changes might explain the lack of editing tolerance for CLK1/2 or the loss of drug efficacy for KKT3.

- The resazurin assay reports metabolic activity rather than cell viability, so it cannot discriminate between dead cells and cells that are only growth-arrested. Therefore, statement in line 277 should be revised. To directly evaluate viability, the authors should complement the resazurin readout with a membrane integrity assay, such as propidium iodide staining, which distinguishes live from nonviable cells.

- The IC50 of the parental strain is quite low, 11.42 uM in promastigote and 19.63 uM in intracellular amastigote (Figure 3), in the absence of naturally sensitive Leishmania kinases, the IC50 should be higher (above 100), as no targets are available. The authors should provide a potential explanation for these findings?

- The authors need to clarify that the values reported lines 287 to 293 are extracted from reference 43, and not derived from Figure 3. This distinction should be made explicit in the text, as presently written, this paragraph is unclear and may mislead readers. In addition, these values should be presented alongside the values obtained from the amastigote results (this study) so that the reader can readily compare the two data sets.

- It should be indicated what data or references underpin the following claim: “possible due to reduced stability of 1NA-PP1 within the acidic environment of the macrophage parasitophorous vacuole” (Line 317-318).

- The authors are encouraged to replace Figure 4a with Supplemental Figure 11a, as the latter presents the data more clearly and is therefore more informative.

- MNC: does this refer to multinucleated cells? The abbreviation is not defined anywhere in the text.

- A clear rationale should be provided for the experimental conditions employed: (i) why a concentration of 5 µM was selected for the assay and why the specific time points of 6 h and 24 h were chosen; and (ii) why, in the parallel experiment using tagged KKT2, a higher concentration of 10 µM was used. Clarifying these decisions will help readers assess the relevance and comparability of the results.

- In Figures 4 and Supplementary Figure 11, the data should be interpreted relative to the treated wild type control rather than to the untreated wild type. At the 6 hour time point (Supplementary Figure 11a) the treated wild type exhibits a reduction of the G₁ population and a concomitant increase in S phase cells, changes that are not observed in the untreated wild type. This shift reveals that 1NM-PP1 itself affects cell cycle progression in the wild type strain, which can alter the interpretation of the KKT2 AS data. For example, at the 24 hour time point, the increase in S phase cells observed in the KKT2 AS is also present in the treated wild type control and is therefore not a phenotype of KKT2 inhibition as stated lines 344 345. Therefore, all subsequent analyses should be compared to the treated wild type condition rather than to the untreated control.

- Figure 4d would benefit from additional microscopy data to substantiate the reported effects on the cell cycle progression of the mutant: (i) images representing cell cycle stages 1 and 5 (also absent from Figure 13) might be included to provide a complete view of the phenotypes observed in the mutant population. Second, corresponding images of wild type cells treated with 1NM PP1 is necessary, because the inhibitor itself impacts the cell cycle; this control will allow direct comparison between treated wild type and mutant cells. Finally, DIC images should be added for each condition so that flagellum duplication can be clearly visualized. Including these elements will give readers a comprehensive and balanced representation of the cell cycle alterations.

- The percentage of cell death at the two time points, 6h and 24h should be provided, to determine the fate of the arrested cells in KKT2-AS treated samples, whether they recover or die.

- The authors should evaluate whether the mNG-tag fused to KKT2 AS alters parasite growth, viability, and cell cycle progression. This assessment should be performed by comparing the KKT2 AS mNG strain with a control line expressing unmodified KKT2 mNG, both in the presence and in the absence of 1NM PP1.

- In Supplementary Figure 14, the 24 hour time point shows a clear rise in MNC together with a concomitant reduction of cells in S phase; however, these changes are not addressed in the text. The authors should discuss these observations and explain their relevance, if any, to the experimental outcomes.

- A clear schematic that illustrates the DMSO versus SM1‑71‑biotin experiment and the subsequent competitive chemoproteomics workflow should be provided. A visual representation of the labelling, enrichment, competition steps, and mass‑spectrometry analysis would greatly aid reader comprehension, as the current description and the Materials‑and‑Methods section are insufficiently detailed.

- In the competition experiment (lines 460 478), 4 µM SM1 71 was used, but, unless I missed it, it does not indicate the concentration of SM1 71 biotin that was added subsequently. The assay is performed on cell lysates, suggesting the absence of washing steps that can remove the SM1 71 and the SM1 71 biotin before the overnight incubation with streptavidin beads. Consequently, competition between the two compounds continues during the pull down, can it affect purification? It is also unclear whether the amount of streptavidin resin used in the protocol is sufficient to capture all PK-bound SM1 71 biotin, considering that the beads could become saturated with free SM1 71 biotin. This could affect enrichment efficiency. The authors should comment on those points.

- The order of the supplementary files is incorrect, Supplementary Data 3 appears before Supplementary Data 2.

PLOS authors have the option to publish the peer review history of their article (what does this mean?). If published, this will include your full peer review and any attached files.

Reviewer #1: No

Reviewer #2: **Yes:** Christian Doerig

Reviewer #3: No

**Figure resubmission:**
---

## [Editor Report · Decision Letter 1]

23 Apr 2026

Dear Dr Brambilla Trindade Carnielli,

We are pleased to inform you that your manuscript 'Chemical genetics reveals Leishmania KKT2 and CRK9 kinase activity is required for cell cycle progression' has been provisionally accepted for publication in PLOS Pathogens.

Best regards,

Michael Boshart

Academic Editor

PLOS Pathogens

Tracey Lamb

Section Editor

PLOS Pathogens

Sumita Bhaduri-McIntosh

Editor-in-Chief

PLOS Pathogens

orcid.org/0000-0003-2946-9497

Michael Malim

Editor-in-Chief

PLOS Pathogens

orcid.org/0000-0002-7699-2064

Dear Juliana, dear Jeremy,

after your response to most of the reviewers´ comments, this very extensive target validation study has gained clarity and further support by the additional controls included. Biochemical analysis of the AS kinases would have been an important addition for mechanistic understanding, but I see that you went great lengths to do this without success. I agree with reviewer 2 that these experiments are not absiolutely essential if not feasible, as the focus of the manuscript is on target validation and you have sufficient in vivo evidence to make the point for two kinases at least.
---

## [Editor Report · Acceptance letter]

Dear Dr B T Carnielli,

We are delighted to inform you that your manuscript, "Chemical genetics reveals Leishmania KKT2 and CRK9 kinase activity is required for cell cycle progression," has been formally accepted for publication in PLOS Pathogens.

Best regards,

Sumita Bhaduri-McIntosh

Editor-in-Chief

PLOS Pathogens

orcid.org/0000-0003-2946-9497

Michael Malim

Editor-in-Chief

PLOS Pathogens

orcid.org/0000-0002-7699-2064